# Convergence of cortical types and functional motifs in the human mesiotemporal lobe

Casey Paquola[1]*, Oualid Benkarim[1], Jordan DeKraker[2], Sara Larivière[1], Stefan Frässle[3], Jessica Royer[1], Shahin Tavakol[1], Sofie Valk[4,5], Andrea Bernasconi[6], Neda Bernasconi[6], Ali Khan[2], Alan C Evans[7,8], Adeel Razi[9], Jonathan Smallwood[10], Boris C Bernhardt[1]*

[1]Multimodal Imaging and Connectome Analysis Lab, McConnell Brain Imaging Centre, Montreal Neurological Institute and Hospital, McGill University, Montreal, Canada; [2]Brain and Mind Institute, University of Western Ontario, London, Canada; [3]Translational Neuromodeling Unit, Institute for Biomedical Engineering, University of Zurich & ETH Zurich, Zurich, Switzerland; [4]Institute of Neuroscience and Medicine (INM-7: Brain and Behaviour), Research Centre Jülich, Jülich, Germany; [5]Institute of Systems Neuroscience, Heinrich Heine University Düsseldorf, Düsseldorf, Germany; [6]Neuroimaging Of Epilepsy Laboratory, McConnell Brain Imaging Centre, Montreal Neurological Institute and Hospital, McGill University, Montreal, Canada; [7]McConnell Brain Imaging Centre, Montreal Neurological Institute and Hospital, McGill University, Montreal, Canada; [8]McGill Centre for Integrative Neuroscience, McGill University, Montreal, Canada; [9]Monash University, Melbourne, Australia; [10]University of York, York, United Kingdom

*For correspondence:
casey.paquola@mcgill.ca (CP);
boris.bernhardt@mcgill.ca (BCB)

Competing interests: The authors declare that no competing interests exist.

**Abstract** The mesiotemporal lobe (MTL) is implicated in many cognitive processes, is compromised in numerous brain disorders, and exhibits a gradual cytoarchitectural transition from six-layered parahippocampal isocortex to three-layered hippocampal allocortex. Leveraging an ultra-high-resolution histological reconstruction of a human brain, our study showed that the dominant axis of MTL cytoarchitectural differentiation follows the iso-to-allocortical transition and depth-specific variations in neuronal density. Projecting the histology-derived MTL model to in-vivo functional MRI, we furthermore determined how its cytoarchitecture underpins its intrinsic effective connectivity and association to large-scale networks. Here, the cytoarchitectural gradient was found to underpin intrinsic effective connectivity of the MTL, but patterns differed along the anterior-posterior axis. Moreover, while the iso-to-allocortical gradient parametrically represented the multiple-demand relative to task-negative networks, anterior-posterior gradients represented transmodal versus unimodal networks. Our findings establish that the combination of micro- and macrostructural features allow the MTL to represent dominant motifs of whole-brain functional organisation.

## Introduction

The mesiotemporal lobe (MTL) is implicated in a diverse range of cognitive processes, notably memory, navigation, socio-affective processing, and higher-order perception (*Moscovitch et al., 2005*; *Squire et al., 2004*; *Zheng et al., 2017*; *Milner, 2005*; *Eichenbaum et al., 2007*; *Wang et al., 2018*; *Eacott et al., 1994*; *Lee and Rudebeck, 2010*; *Buzsáki and Moser, 2013*; *Felix-Ortiz and Tye, 2014*; *Lech and Suchan, 2013*). Contemporary views recognise that the broad role this region

plays in human cognition emerges through interactions between neural codes within the MTL and the rest of the brain (*Lech and Suchan, 2013*; *Saksida and Bussey, 2010*; *Staresina and Davachi, 2009*; *Diana et al., 2007*). Motivated by its role in spatial processing (*O'Keefe and Nadel, 1978*), it has been suggested that the MTL plays a key role in the formation of cognitive maps (*Tolman, 1948*) by providing a low dimensional representational scheme of different situational contexts (*O'Keefe and Nadel, 1978*; *Behrens et al., 2018*). A widely held assumption posits that the broad role of the MTL in human cognition results from both its intrinsic circuitry and position within the larger cortical architecture, which places the MTL at the apex of multiple conceptual hierarchies (*Mesulam, 1998*; *Felleman and Van Essen, 1991*; *Chanes and Barrett, 2016*).

Unique in the mammalian brain, the mesiotemporal cortex folds in upon itself, producing the curled hippocampal formation that sits atop the parahippocampal region. This infolding is accompanied by a shift from six-layered isocortex to three-layered allocortex (*Insausti et al., 2017*). During ontogeny, isocortex (from iso- 'equal') differs from allocortical (from allo- 'other') hippocampus, which lacks clear lamination prenatally and exhibits a more even distribution of neurons across depths (*Sanides, 1962*; *Vogt and Vogt, 1919*). Such variations in cytoarchitecture are thought to be involved in the transformation from unimodal to amodal information (*Lavenex and Amaral, 2000*), as well as degree of synaptic plasticity low in iso-, and high in allocortex (*García-Cabezas et al., 2017*). The transition from allo-to-isocortex occurs in smooth streams of gradation emanating from the hippocampus (*Sanides, 1962*; *Duvernoy et al., 2013*; *Braak and Braak, 1985*). This wave of laminar differentiation is phylogenetically conserved (*Sanides, 1962*; *Dart, 1934*; *Abbie, 1938*; *Abbie, 1942*) and classic histological studies suggested that the wave is accompanied by step-wise changes in neuronal density at various cortical depths (*Sanides, 1969*). Nevertheless, compared to other mammals, the human MTL has exaggerated folding, more extensive lamination of the entorhinal cortex as well as a more prominent appearance of the Cornu Ammonis (CA) 2 subfield, highlighting the need to characterise the region's microstructure in our species (*Insausti et al., 2017*; *Duvernoy et al., 2013*; *Insausti and Amaral, 2012*). Previous studies have detailed the distinct cytoarchitectural properties of hippocampal subfields (*DeKraker et al., 2019*), but the cytoarchitectural patterns that extend across the subfields and their relation to the isocortex are less clear.

Substantial cytoarchitectural differentiation within the MTL holds important implications for signal flow. Degree of laminar differentiation is generally thought to reflect the origin and termination patterns of axonal projections (*Barbas, 1986*), and so the connectivity between cytoarchitecturally distinct MTL subregions may thus be described as feedforward or feedback based on relative iso-to-allocortical position. Accordingly, feedforward projections flow from iso-to-allocortex, whereas feedback projections flow from allo-to-isocortex (*Barbas, 1986*; *Hilgetag and Grant, 2010*). Thus, mapping directional signal flow along the axis could inform upon the role of the MTL in feedforward or feedback processing. Furthermore, cytoarchitectural similarity is tightly coupled with likelihood of interareal connectivity, a principle known as the 'structural model' (*Barbas, 1986*; *Beul et al., 2015*; *Beul et al., 2017*; *García-Cabezas et al., 2019*), suggesting that the intrinsic circuitry of the MTL is guided by the iso-to-allocortical axis. Consistent with this view, key fibre pathways in the MTL, notably the perforant pathway, mossy fibres, and Schaffer collaterals, are typically defined by their relation to hippocampal subfields (subiculum, CA1-4, dentate gyrus) that are sequentially organised along the iso-to-allocortical axis (*Amaral and Witter, 1989*). In addition to the iso-to-allocortical shifts that follow hippocampal infolding, neurobiological and functional properties of the MTL system also appear to be organised with respect to a second, anterior-posterior axis (*Amaral and Witter, 1989*; *Strange et al., 2014*; *Witter et al., 2006*; *Poppenk et al., 2013*; *Vogel et al., 2020*). Tract-tracing studies in rodents have shown that anterior-posterior gradients determine hippocampal connectivity to entorhinal cortices (*Witter et al., 2006*) and functional neuroimaging studies in humans have illustrated distinct combinations of anterior-posterior and lateral-medial topographies in the entorhinal and parahippocampal cortex (*Navarro Schröder et al., 2015*; *Maass et al., 2015*). Recently, data-driven analyses of resting state functional magnetic resonance imaging (MRI) have confirmed marked differentiation of anterior and posterior aspects of the hippocampus in terms of intrinsic functional connectivity (*Zhong et al., 2019*; *Vos de Wael et al., 2018*; *Przeździk et al., 2019*).

Given the established role of the MTL as a hub of multiple large-scale networks (*Smallwood et al., 2016*; *Vincent et al., 2006*; *Ojemann et al., 1997*; *Braga and Buckner, 2017*), the combination of both iso-to-allocortical and anterior-posterior axes may provide insight into its'

broad contribution to neural function and different cognitive states. This architecture could allow operations implemented by the internal hippocampal circuitry, such as pattern separation and completion (*Rolls, 2016*; *Neunuebel and Knierim, 2014*; *Staresina et al., 2016*; *Nakazawa et al., 2002*), to be realised across multiple cortical processing zones in a distributed and coordinated manner. In this way, the coarse representations in the MTL could be mirrored by neural motifs present across the broader cortical system, helping to explain the regions' contribution to multiple aspects of memory (*Clark, 2018*), and to cognitive maps describing the structure of many different tasks (*Eichenbaum and Cohen, 2014*; *Solomon et al., 2019*; *Tavares et al., 2015*). Contemporary neuroscience has established that whole-brain functional organisation can be understood as discrete communities such as the sensory, attention, and transmodal networks (*Yeo et al., 2011*) and that the relationships between these communities can be represented by gradients of cortex-wide connectivity (*Margulies et al., 2016*). Critically, these approaches highlight overlapping neural motifs that are thought to play an important role in different features of cognitive function. One functional gradient differentiates sensory from transmodal systems that harbour cognitive processes increasingly reliant on information from memory (*Murphy et al., 2018*; *Murphy et al., 2019*; *Sormaz et al., 2018*). A second large-scale functional gradient reflects the dominance of networks involved in complex tasks in which there is no predetermined schema with which to guide behaviour, also referred to as the multi demand system (*Fedorenko et al., 2013*; *Duncan, 2010*). Extensive research on preferential connectivity of anterior hippocampus to transmodal isocortex and posterior hippocampus to sensory isocortex shows that the MTL architecture can reflect a large-scale functional motif (*Strange et al., 2014*; *Maass et al., 2015*; *Vos de Wael et al., 2018*; *Przeździk et al., 2019*; *Libby et al., 2012*). Our study explored whether the iso-to-allocortical axis also reflects a large-scale functional motif, and whether the axis interactions could help understand how the MTL elicits a broad role in neural function.

We leveraged an ultra-high-resolution 3D reconstruction of a histologically stained and sliced human brain ( *BigBrain* [*Amunts et al., 2013*]) to build a continuous model of the MTL architecture. Surface-based approaches were used to sample intracortical microstructure across multiple cortical depths along both the iso-to-allocortical and anterior-posterior axes. We hypothesised that the iso-to-allocortical axis would be the principle gradient of cytoarchitectural variation in the MTL and identified salient cytoarchitectural signatures of this transition using supervised machine learning. We then translated the histology-based model of the MTL to in-vivo functional neuroimaging to assess how the iso-to-allocortical and anterior-posterior axes covary with intrinsic and large-scale network connectivity. Building upon established models of internal hippocampal circuitry (*Amaral and Witter, 1989*), we hypothesised that the strongest signal flow throughout the MTL would follow the iso-to-allocortical axis, and tested directional signal flow via dynamic causal modelling of resting state functional MRI (*Friston et al., 2014*). In addition to testing consistency of these intrinsic circuit characteristics across the anterior-posterior axis, we assessed how both the iso-to-allocortical and anterior-posterior axes relate to macroscale functional organisation by assessing the topographic representation of large-scale functional gradients in the MTL space.

## Results

### Overview

We developed a detailed model of the confluence of cortical types in the parahippocampus-hippocampus complex based on an 40 μm 3D histological reconstruction of a human brain (see Materials and methods; *Figure 1A*), which we then translated to in-vivo functional imaging and macroscale connectomics. Considering MTL cytoarchitecture, we observed a strong gradient along the iso-to-allocortical axis, with depth-wise intracortical profiles acting as robust predictors of spatial axis location. The most predictive feature was consistent across the anterior-to-posterior regions, suggesting a preserved iso-to-allocortical architecture along the MTL long axis. Mapping the cortical confluence model to in-vivo resting state functional MRI, intrinsic signal flow was strong along the iso-to-allocortical axis but interacts with the long-axis. We found that the iso-to-allocortical and anterior-to-posterior axes differentially contributed to distinct dimensions of macroscale functional systems, with anterior-to-posterior position reflecting trade-offs between sensory and transmodal

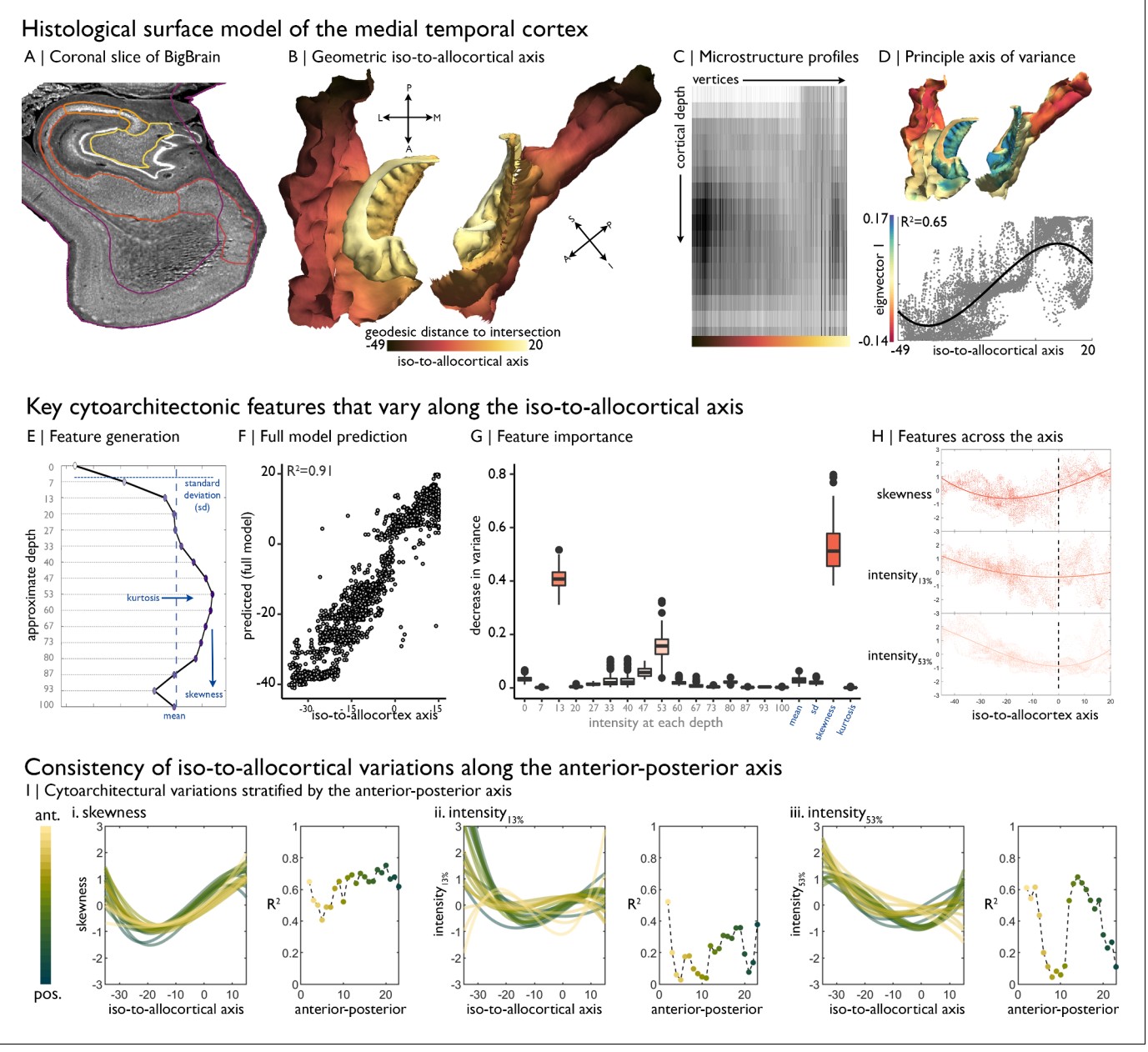

**Figure 1.** Cytoarchitectural profiling of the mesiotemporal confluence. (**A**) Isocortical and allocortical surfaces projected onto the 40 µm *BigBrain* volume. Notably, conventional isocortical surface construction skips over the allocortex. Purple = isocortex. Red = subiculum. Dark orange = CA1. Orange = CA2. Light orange = CA3. Yellow = CA4. (**B**) The iso-to-allocortical axis was estimated as the minimum geodesic distance to the intersection of the isocortical and hippocampal surface models. (**C**) Intensity sampling along 16 surfaces from the inner/pial to the outer/white surfaces produced microstructure profiles. Darker tones represent higher cellular density/soma size. (**D**) *Above.* The principle eigenvector/axis of cytoarchitectural differentiation, projected onto the confluent surface. *Below.* The association between the principle eigenvector and the iso-to-allocortical axis, with the optimal polynomial line of best fit. (**E**) Cytoarchitectural features generated for each microstructure profile. (**F**) Empirical *vs* predicted position on the iso-to-allocortical, based on supervised random forest regression with cross-validation. (**G**) Feature importance was approximated as how much the feature decreased variance in the random forest. (**H**) Cubic fit of each selected feature with the iso-to-allocortical axis. Feature values are z-standardised. (**I**) Line plots show the cubic fit of cytoarchitectural features to the iso-to-allocortical axis within 23 bins of the anterior-posterior axis (*yellow-to-green*). Neighbouring scatter plots depict the goodness of fit (adjusted $R^2$) of each polynomial, showing high consistency of the pattern of skewness.

The online version of this article includes the following figure supplement(s) for figure 1:

**Figure supplement 1.** Manifold learning of MTL cytoarchitecture and relation to subregions.

**Figure supplement 2.** Exemplar microstructure profiles of selected features, namely skewness, intensity at ~13% depth and ~53% depth.

function, and iso-to-allocortical positions related to the relative contribution of the multiple-demand system.

## Cytoarchitectural analysis of the iso-to-allocortical confluence

We first mapped the iso-to-allocortical axis as the minimum geodesic distance of a vertex to the intersection of the isocortical and allocortical surface models (*Figure 1B*). To examine the cytoarchitecture of the cortical confluence, we generated microstructure profiles, reflecting staining intensity across 16 cortical depths (*Figure 1C*). Unsupervised manifold learning of microstructure profile data revealed that the principle gradient of cytoarchitectural differentiation, accounting for 15.6% of variance, closely approximates the geometric iso-to-allocortical axis (r = 0.76, p<0.001; *Figure 1D*). This relationship appeared nonlinear (adjusted $R^2$ for polynomials 1–3 = 0.58/0.59/0.65), and CA2 expressed a different cytoarchitecture than expected by its position on the geometric map (*Figure 1—figure supplement 1*). Lower eigenvectors were not spatially correlated with the iso-to-allocortical axis (0.01<|r| < 0.31), demonstrating specificity of the first eigenvector in mapping this cytoarchitectural gradient (see *Figure 1—figure supplement 1*, for lower eigenvectors).

Microstructure profile features provided a highly accurate mapping of the iso-to-allocortical axis based on random forest regression ($R^2$ = 0.91 ± 0.005; *Figure 1E–F*). Profile skewness, as well as staining intensity at upper (~13% depth) and mid (~53% depth) surfaces were key features in learning the axis (average reduction in variance: 53/41/16%; *Figure 1G*). These three features alone accounted for most variance in out-of-sample data ($R^2$ = 0.86 ± 0.006; *Figure 1—figure supplement 2*). Microstructure profile skewness increased from iso-to-allocortex, which pertained to a shift from a profile with mid-depth peak to a flatter microstructure profile (*Figure 1H*, see *Figure 1—figure supplement 2* for exemplar microstructure profiles and feature values). Intensity at ~13% depth approximately corresponds to the layer1/2 boundary in the isocortex and exhibited a rapid uptick at the iso-allocortical intersection. Intensity at ~53% depth approximately aligns with the peak intensity of the average microstructure profile, signifying high cellular density, and reached a minimum around the intersection of the iso- and allo-cortex.

Next, we sought to test whether the iso-to-allocortical variations were consistently expressed along the anterior-posterior axis. Independently inspecting 1 mm coronal slices of the *post mortem* data, we found that microstructure profile skewness consistently increased along iso-to-allocortical axis, regardless of anterior-posterior axis (mean ± SD adjusted $R^2$ = 0.64 ± 0.17, *Figure 1I left*). On the other hand, depth-dependent intensities were more variable along the anterior-posterior axis (13% depth: goodness of fit = 0.20 ± 0.12; 53% depth: goodness of fit = 0.40 ± 0.24) and exhibited different iso-to-allocortical patterns in the anterior portion. Thus, while certain depth-dependent intensities systematically varied across both geometric axes, the most salient cytoarchitectural feature (skewness) consistently and gradually decreased across the iso-to-allocortical axis regardless of long-axis position.

## Iso-to-allocortical gradients in internal signalling

Functional signal flow within axes of the mesiotemporal confluence was examined via dynamic modelling of resting state functional MRI, using a generative model of effective connectivity, after transforming the cortical confluence model first from histological to stereotaxic MNI152 space (*Figure 2A*), and then to native functional MRI spaces in a group of 40 healthy adults (see Materials and methods for demographics, acquisition and preprocessing). In brief, we extracted blood oxygen level dependent timeseries from each voxel of the cortical confluence during a resting state scan, averaged timeseries within a discrete set of bins of the iso-to-allocortical axis and carried out a Bayesian model reduction of a dynamic causal model.

This model showed dense, bidirectional effective connectivity across the MTL (*Figure 2B*). To test whether effective connectivity predominantly followed the iso-to-allocortical axis, and if so in which direction, we labelled edges by deviation from the axis, whereby lower deviation represented an edge more concordant with the iso-to-allocortical axis (*Figure 2B far right*). Deviation from the iso-to-allocortical axis was related to lower effective connectivity in both directions (to isocortex $r_{iso}$ = −0.43, p=0.02; to allocortex: $r_{allo}$ = −0.44, p=0.02; difference in coefficients: z = −4.33, p=0.99, *Figure 2B right*). Furthermore, the final bin, containing CA4, inhibited other regions and the

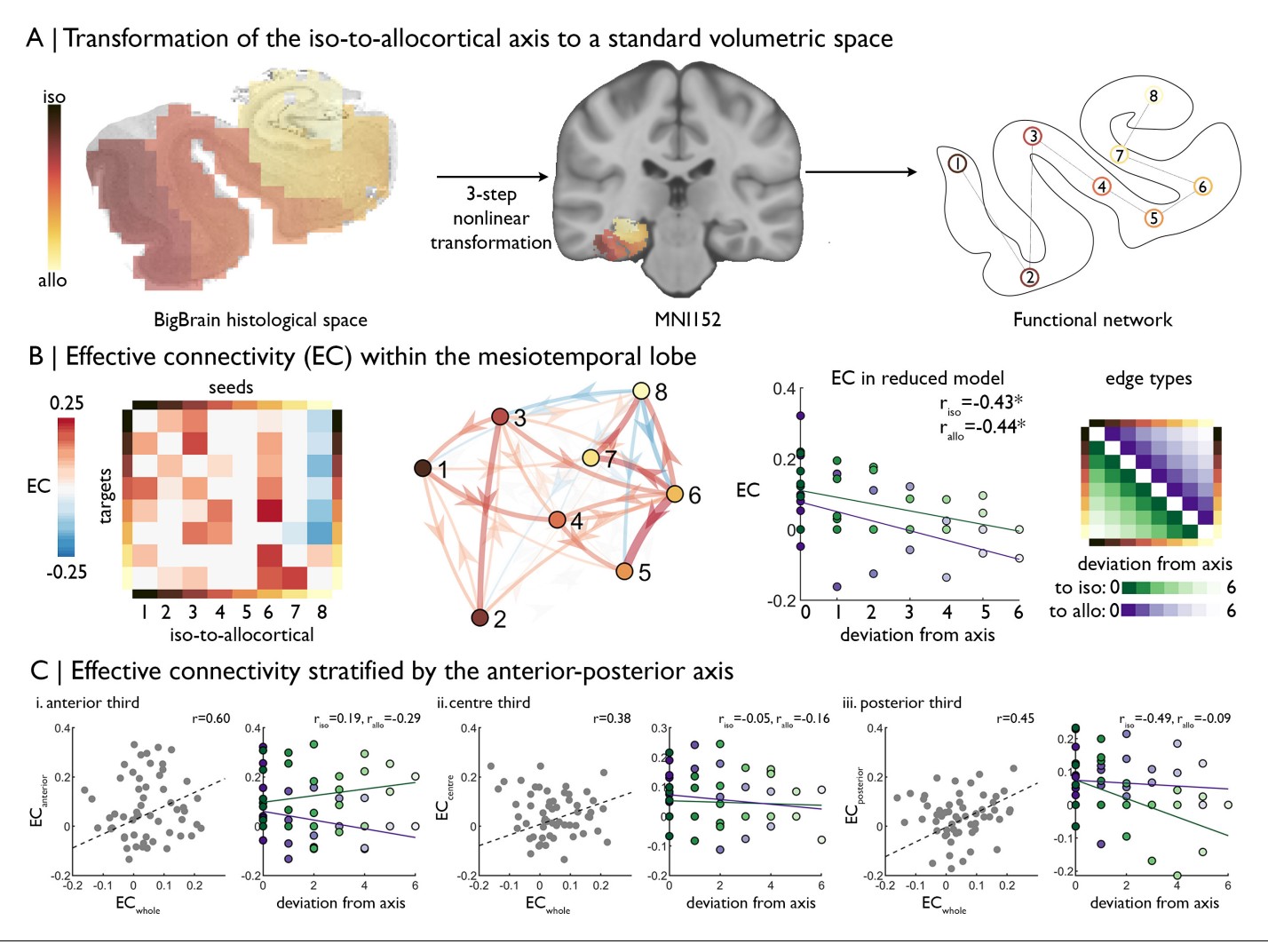

**Figure 2.** | Intrinsic functional signal flow across the cortical confluence. (**A**) The surface-based iso-to-allocortical axis was projected into the native volume space of *BigBrain*, then registered to stereotaxic MNI152 space. We used this as a standard atlas to extract resting state functional MRI timeseries. An 8-node decomposition was used for the dynamic model (subregion overlap shown in ***Figure 2—figure supplement 1*** ). (**B**) *Left* Posterior estimates of effective connectivity (EC) between 8-nodes of the iso-to-allocortical axis. Columns are seeds and rows are targets. *Right.* Strength of effective connectivity with deviation from the iso-to-allocortical axis, stratified by direction. * signifies $p_{FDR}$ <0.05. (**C**) Association of effective connectivity within anterior-posterior thirds of cortical confluence model (y-axes) with effective connectivity of the whole model and deviation from the iso-to-allocortical axis (x-axes).

The online version of this article includes the following figure supplement(s) for figure 2:

**Figure supplement 1.** Alignment with subfields and functional homogeneity of iso-to-allocortical bins.

sixth bin, corresponding to the subiculum, was the strongest driver of excitatory effective connectivity (see ***Figure 2—figure supplement 1*** for subfield overlap).

Next, we tested the interaction of iso-to-allocortical and anterior-posterior axes in determining intrinsic signal flow by constructing the dynamic model within thirds of the anterior-posterior axis. If only the iso-to-allocortical axis determines the intrinsic connectivity, then the pattern of effective should be conserved within each third. Effective connectivity estimates, calculated from a model within thirds of the anterior-posterior axis were moderately correlated with effective connectivity estimates from the full mesiotemporal confluence model (r = 0.60/0.38/0.45, ***Figure 2C***). Unlike the full mesiotemporal confluence, however, we did not consistently observe inverse correlation between deviation from the iso-to-allocortical axis and effective connectivity in these spatially

restricted models (difference in coefficients: anterior z = −2.92, p=0.99; middle z = −1.01, p=0.84; posterior z = −0.47, p=0.68, *Figure 2C*).

### Iso-to-allocortical and anterior-posterior axes represent macroscale functional organisation

After demonstrating subtle interactions of iso-to-allocortical and anterior-posterior axes in cytoarchitectural and intrinsic signalling of the MTL, we examined the interplay of both axes in the topographic representation of macroscale functional systems within the MTL. To this end, we examined conventional rs-fMRI connectivity between MTL voxels and isocortical parcels, with large-scale networks being characterised with respect to both previously established canonical functional communities (*Yeo et al., 2011*) and macroscale connectome gradients (*Margulies et al., 2016*; *Figure 3A–C*).

The strength of rs-fMRI connectivity between the MTL and the isocortex varied as a function of the iso-to-allocortical axis. Specifically, lateral dorsal attention and fronto-parietal networks exhibited stronger rs-fMRI connectivity towards the isocortical anchor of the cortical confluence, whereas posterior cingulate and medial prefrontal regions of the default-mode network and medial occipital areas in the visual network showed stronger connectivity towards the allocortical anchor (*Figure 3D*). A similar pattern was summarized by observing a parametric relation between position on the iso-to-allocortical axis and rs-fMRI along the third functional gradient (r = −0.74, $p_{spin}$ <0.001), which depicts a differentiation of the multiple-demand system (*Fedorenko et al., 2013*; *Duncan, 2010*). The multiple-demand anchor of the third functional gradient had stronger connectivity to isocortical compartments of the mesiotemporal confluence, whereas the opposing anchor comprising transmodal and sensory networks had higher connectivity with allocortical compartments. The association appears to be specific to the third functional gradient, supported by higher correlation coefficients than the first (z = 12.30 p<0.0001) and second functional gradients (z = 21.33 p<0.0001).

While we found that the iso-to-allocortical representation of the multiple-demand gradient was preserved within thirds of the long-axis (correlation of r map with G3: 0.27 < r < 0.40), long-axis position nevertheless affected representation of macroscale topographies specifically with respect to the first, sensory-transmodal functional gradient (correlation of r map with G1: −0.41 < r < 0.45). Further elucidating preferential rs-fMRI connectivity patterns of the anterior-posterior axis, we indeed found that this largely reflected connectivity transitions described by the sensory-transmodal functional gradient (r = −0.84, $p_{spin}$ <0.001; *Figure 3E*). The transmodal anchor had stronger functional connectivity with the anterior aspect of the MTL, while the sensory anchor had stronger connectivity with the posterior aspect. Again, the association was specific to one gradient, with higher correlation coefficients of the anterior-posterior axis with the first functional gradient than the second (z = 19.21, p<0.001) and third (z = 30.23, p<0.001).

Although our included datasets have overall high data quality, signal drop out in temporal lobe areas may potentially affect imaging findings. To dispel such concerns, we assessed signal to noise ratio (SNR) from the rs-fMRI data. While we indeed observed variations in both temporal as well as spatial SNR across the MTL, findings were virtually identical controlling for SNR via partial correlation [compared to the iso-to-allocortical axis from *Figure 3B*: $r_{spatial}$ = 0.92, $r_{temporal}$ = 0.99; compared to the anterior-posterior axis *from Figure 3C*: $r_{spatial}$ = 0.99, $r_{temporal}$ = 0.99]. Similarly, functional findings from mesiotemporal confluence were virtually consistent when excluding of temporal lobe regions as targets in the macrolevel functional connectivity findings (compared to *Figure 3F*: r = 0.85).

Together, these analyses demonstrate an important interaction of the iso-to-allocortical and anterior-posterior axes of the MTL in understanding its functional relationship to the rest of the brain, with shifts in the iso-to-allo-cortical axis relating to connectivity to the multiple-demand gradient and shifts along anterior-to-posterior axis relating to connectivity in the sensory-transmodal gradient (*Figure 3F*).

## Replication of functional analyses

While the singular nature of *BigBrain* did not allow for additional cytoarchitectural cross-validations, we could test consistency of our in-vivo functional analyses in an independent multi-modal imaging

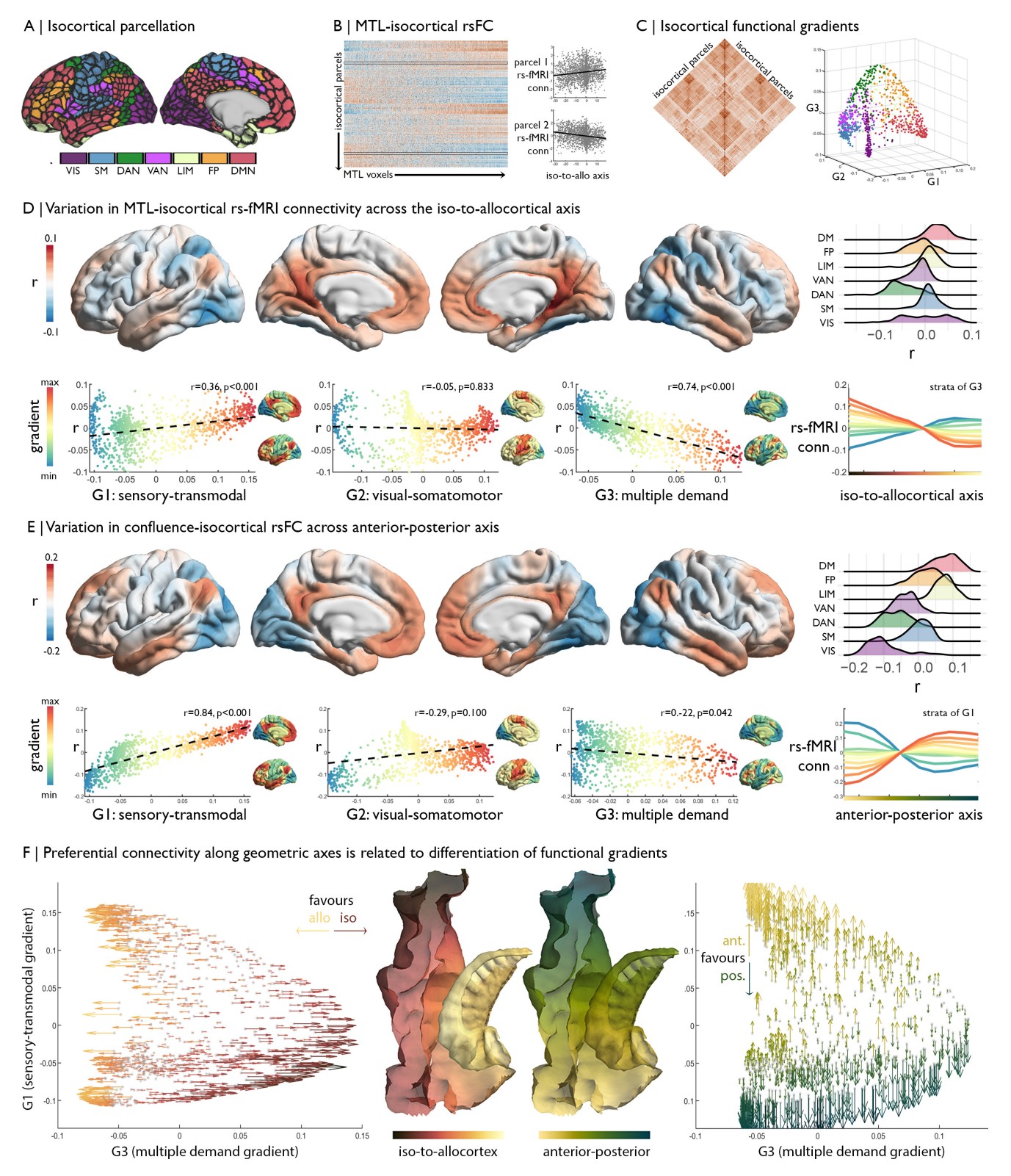

**Figure 3.** Relation of MTL axes with large-scale functional motifs. (**A**) Parcellation of the isocortex, coloured by functional community (*Schaefer et al., 2018*). (**B**) rs-fMRI connectivity between isocortical parcels and MTL voxels, with examples of row-wise associations with the iso-to-allocortical axis. (**C**) Isocortical rs-fMRI connectivity was decomposed into a set of functional gradients, shown in a 3D space on the right. (**D**) Surface maps show parcel-wise correlation of rs-fMRI connectivity with iso-to-allocortical axis. Red indicates increasing connectivity towards the allocortex, whereas blue indicates

*Figure 3 continued on next page*

*Figure 3 continued*

increasing functional connectivity towards the isocortex. The ridgeplot depicts the r values for each community. Scatter plots depict how regional associations with iso-to-allocortical axis (r, y-axis) vary as a function of the position each of the functional gradients (x-axis). Functional gradients are presented on the cortical surface insets right to each scatter plot. Line plot shows the average rs-fMRI connectivity ten strata of the third functional gradient with MTL voxels, organised along the iso-to-allocortical axis. (E) We repeated the analysis in D for the anterior-posterior axis, showing a strong association with the sensory-transmodal gradient. (F) Both scatterplots depict isocortical parcels in functional gradient space, defined by the sensory-transmodal and the multiple-demand gradients. Arrows represent the correlation of rs-fMRI connectivity from that isocortical parcel with the iso-to-allocortical (*left*) and the anterior-posterior axes (*right*). Arrow direction reflects the sign of the correlation, corresponding to the respective axis, and the colour depicts the strength.

dataset (Human Connectome Project; see Materials and methods for details). In line with the original analysis, we observed decreasing effective connectivity with increasing deviation from the iso-to-allocortical axis ($r_{iso}$ = −0.24; $r_{allo}$ = −0.47; *Figure 4A*), meaning there was stronger effective connectivity between closer neighbours on the iso-to-allocortical axis than to off axis nodes. Effects varied within anterior, middle and posterior thirds (*Figure 4A* right), again demonstrating intrinsic signal flow over iso-to-allocortical and anterior-posterior axes. Additionally, the iso-to-allocortical and anterior-posterior axes reflected distinct macroscale functional topographies. As in the original analyses, strength of isocortical rs-fMRI connectivity with the iso-to-allocortical axis corresponded with the multiple-demand functional gradient (r = −0.52, $p_{spin}$ <0.001), whereas variations in rs-fMRI connectivity along the anterior-posterior axis corresponded with the sensory-transmodal functional gradient (r = 0.33, $p_{spin}$ <0.001) (*Figure 4B*).

## Discussion

Our study set out to determine whether macro and microstructural features of the human mesiotemporal lobe (MTL) offers an explanation for its broad influence on neural function. The MTL, a cytoarchitecturally unique region that harbours a transition from three-to-six-layered cortex, has been a key focus of neuroscience for decades. Its intrinsic circuity is generally considered a model system to study how structural and functional properties co-vary in space (*Cembrowski et al., 2018*; *Cembrowski et al., 2016*; *Masukawa et al., 1982*). Furthermore, the MTL is known to have distributed connectivity patterns to multiple macroscale brain networks and to participate in numerous cognitive processes (*Moscovitch et al., 2005*; *Squire et al., 2004*; *Milner, 2005*; *Eichenbaum et al., 2007*; *Wang et al., 2018*; *Eacott et al., 1994*; *Lee and Rudebeck, 2010*; *Buzsáki and Moser, 2013*; *Felix-Ortiz and Tye, 2014*; *Wang et al., 2016*). Our work suggests that the intersection of the iso-to-allocortical cytoarchitectural gradient with the anterior-posterior axis allows the MTL to topographically represent multiple, dominant motifs of whole-brain functional organisation, which facilitates the distribution of neural codes computed within the internal hippocampal circuit across the putative cortical hierarchy. Our starting point was an observer-independent characterisation of cytoarchitectural transitions in the MTL. To this end, we integrated manual hippocampal segmentation (*DeKraker et al., 2019*) with an isocortical surface to create a smooth, continuous surface model on an ultra-high-resolution 3D histological reconstruction of the human brain (*Amunts et al., 2013*). This approach allowed for a depth-wise microstructure profiling along a continuous geometric axis running through the folds of the MTL. We characterised microstructure profiles by the central moments, shown to discriminate between specific MTL segments (*Insausti et al., 2017*; *Duvernoy et al., 2013*; *Insausti and Amaral, 2012*) as well as depth-wise intensities that provide a complementary and more direct interpretation. Several analyses specifically interrogated the transition from parahippocampal isocortex with a six-layered structure to three-layered hippocampal allocortex. A data-driven framework, based on unsupervised manifold learning of microstructure profile covariance (*Paquola et al., 2019b*), revealed a smooth and continuous iso-to-allocortical gradient as the principle axis of variation in cytoarchitecture in the MTL. Although our approach did not specifically aim at decomposing the MTL into discrete subfields with sharp boundaries (see [*de Flores et al., 2020*; *Wisse et al., 2017*] for histological and MRI efforts), the subfields of the MTL were nevertheless embedded within the iso-to-allocortical gradient and transitions from one subfield to the next were relatively smooth. One exception was CA2, however, which is recognized for high neuronal density (*DeKraker et al., 2019*) and unique synaptic properties within the

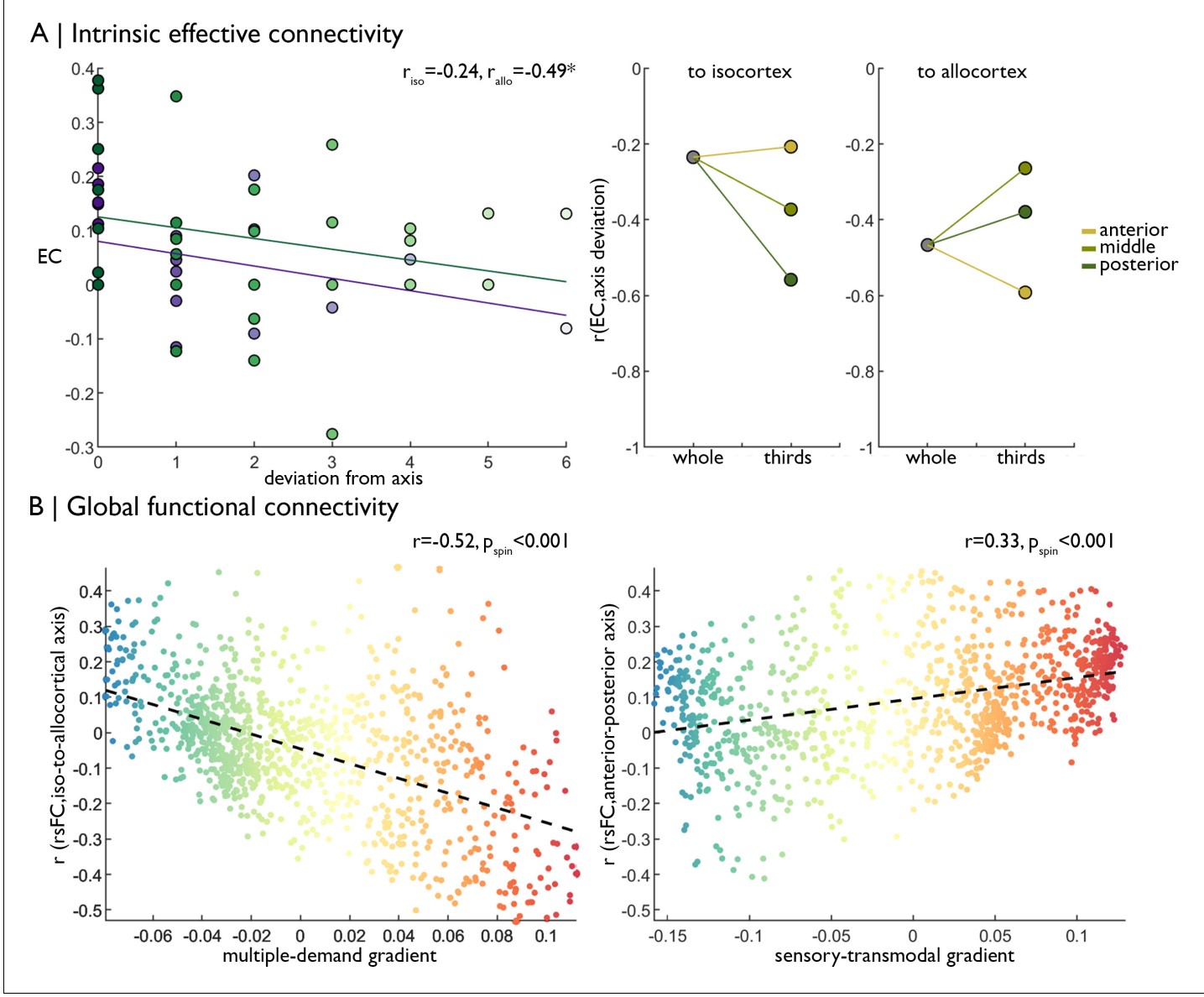

**Figure 4.** Replication of functional analyses in an independent dataset. (**A**) *Left* Strength of effective connectivity decreases with deviation from the iso-to-allocortical axis. Green = to isocortex. Purple = to allocortex. *Right* The correlation between effective connectivity and deviation from iso-to-allocortical axis differed within anterior, middle and posterior thirds. (**B**) Scatterplots show relationship between varying strength of rs-fMRI along the iso-to-allocortical (*left*) and anterior-posterior (*right*) axes mirror multiple-demand and sensory-transmodal isocortical functional gradients, respectively.

larger CA complex (*Dudek et al., 2016*; *Lorente de Nó, 1934*). Manifold learning was complemented by quantitative approaches to fingerprint cytoarchitecture based on statistical moments (*Zilles et al., 2002*; *Paquola et al., 2019a*; *Schleicher et al., 1999*) and an evaluation of their accuracy in predicting location along the iso-to-allocortical axis for a given intracortical profile. Here, the a key feature to predict the iso-to-allocortical transition in our model was skewness. Profile skewness, reflecting relative staining intensity across cortical depths, captured the expected shift from a peaked isocortical profile to a flatter allocortical microstructure profile. The uppermost molecular layer, sparsely populated by neurons in isocortex, appears less prominent towards the three-layered allocortex (*Insausti et al., 2017*), an effect that is likely accentuated by decreasing cortical thickness (*Insausti et al., 2017*). The pattern was conserved along the anterior-posterior axis, showing that the skewness-based cytoarchitectural gradient holds similar influence along the long axis. Finally, this

iso-to-allocortical cytoarchitectural gradient extends upon the principle axis of microstructural differentiation identified across the isocortex (*Paquola et al., 2019a*). Together, this work shows a gradual differentiation in cyto- and myelo-architecture from sensory to limbic areas defines a core organisational axis in humans, extending from prior work in non-human primates and rodents (*Paquola et al., 2019b*; *Paquola et al., 2019a*; *Burt et al., 2018*; *Fulcher et al., 2019*; *Huntenburg et al., 2018*; *Huntenburg et al., 2017*).

As the next step, we assessed whether the cytoarchitectural gradient along the iso-to-allocortical axis confers the intrinsic functional organisation of the MTL, complementing previous modular descriptions of its internal circuitry that posit dense connectivity between parahippocampal, perirhinal/entorhinal, and hippocampal subregions (*Lacy and Stark, 2012*; *Shah et al., 2018*). Projecting the confluence model to in-vivo MRI and studying estimates of directed connectivity, the evidence supported a dominant flow of connectivity along the iso-to-allocortical axis, in line with the purported link between cytoarchitectural similarity and inter-regional connectivity (*Barbas, 1986*; *Hilgetag and Grant, 2010*; *Beul et al., 2017*; *Barbas and Pandya, 1989*; *Barbas and Rempel-Clower, 1997*). Signal flow along iso-to-allocortical axis is known to be mediated by intracortical fibre tracts along the MTL infolding, notably mossy fibres and Schaefer collaterals (*Amaral and Witter, 1989*; *Zeineh et al., 2017*). While signal flow and cytoarchitectural measurements are inextricably linked to distance, the diminishment of this pattern within anterior-posterior thirds of the MTL shows that it is not purely a product of spatial proximity. Following the relative preservation of the iso-to-allocortical cytoarchitectural gradient along the MTL long axis, signal flow patterns were also statistically similar along different anterior-to-posterior segments. Yet, we nevertheless observed subtle differences, supporting interactions between anterior-posterior and iso-to-allocortical axes on the internal MTL circuitry. This effect is likely driven by the variable connectivity of the hippocampus to the surrounding isocortex along the anterior-posterior axis, rather than anterior-posterior variations in intrinsic connectivity of the hippocampus proper. Projections from the parahippocampus to the entorhinal cortex, subiculum and CA1 are clearly stratified into anterior (perirhinal) and posterior (postrhinal) tracts in the rodent brain (*Naber et al., 1999*; *Naber et al., 2001*), which converge as early as the lateral entorhinal cortex (*Doan et al., 2019*). In line with this evidence, recent neuroimaging studies have shown more variability in rs-fMRI connectivity between anterior-posterior aspects of the MTL than between hippocampal subfields (*Dalton et al., 2019*) and that subregions of the parahippocampal and entorhinal cortex exhibit preferential connectivity to either anterior or posterior aspects of the subiculum (*Maass et al., 2015*). Together, this evidence demonstrates that distinct anterior and posterior input streams to the hippocampus are integrated and flow along the iso-to-allocortical cytoarchitectural gradient. In other words, the steep descent in laminar differentiation across the cortical confluence was found to reflect signal flow along the iso-to-allocortical axis and pulls together the anterior and posterior processing streams within the MTL.

We furthermore characterised the position of the MTL in the macroscale cortical landscape, using both modular and dimensional decompositions of whole-brain function (*Yeo et al., 2011*; *Margulies et al., 2016*; *Schaefer et al., 2018*; *Vos de Wael et al., 2020*). Particularly the latter framework has recently been employed to describe variations in microcircuitry (*Fulcher et al., 2019*; *Chaudhuri et al., 2015*; *Burt, 2017*), large-scale functional organisation (*Mesulam, 1998*), and putative neural hierarchies (*Chanes and Barrett, 2016*). Our work demonstrated that the two major geometric axes of the MTL exhibited complementary affiliation to macroscale cortical systems and thereby balance distinct modes of cortical dynamics. The iso-to-allocortical axis is related to decreasing connectivity with multiple-demand systems - multimodal areas involved in external attention across multiple contexts (*Fedorenko et al., 2013*; *Duncan, 2010*) - while the anterior-posterior axis reflected a shift from preference for transmodal cortex, such as the default-mode network, to sensory-and-motor regions. Our results thus suggest that the anterior and posterior processing streams are successively integrated along steps of the iso-to-allocortical axis while external attentional processes are increasingly dampened. Higher-order representations constructed within the depths of the hippocampus then flow from allo-to-isocortex before being broadcast back to sensory and transmodal areas. This 'hierarchy of associativity' is also evidenced by invasive tract-tracing in non-human animals (*Lavenex and Amaral, 2000*) and could provide a neural workspace that is capable of representing both the abstract structure of tasks, as well as their specific sensory features, thereby supporting the hypothesised role of the MTL in cognitive maps (*O'Keefe and Nadel, 1978*; *Tolman, 1948*; *Behrens et al., 2018*). To our knowledge, no study to date has specifically examined

continuous variations in functional connectivity along the iso-to-allocortical axis of the MTL; however, our results align with observed participation of the parahippocampus and fusiform gyri in externally-oriented and attention networks (*Yeo et al., 2011*; *Qin et al., 2016*; *Andrews-Hanna et al., 2010*). The iso-to-allocortical axis also mirrors a reduction in the diversity of projections (*Lavenex and Amaral, 2000*), which further explains the shift from involvement in large-scale polysynaptic networks towards more constrained, local connectivity. These findings may help to understand the influence of cytoarchitecture and connectivity on functional dissociation within the MTL, as subfield-focused approaches have shown selective task activation along the iso-to-allocortical axis, for example during pattern separation/completion (*Stevenson et al., 2020*) and encoding/retrieval (*Suthana et al., 2015*). Sensory-transmodal differentiation of rs-fMRI connectivity along the MTL long axis is well-evidenced in the literature (*Strange et al., 2014*; *Maass et al., 2015*; *Vos de Wael et al., 2018*; *Przeździk et al., 2019*; *Libby et al., 2012*) and indicates more anatomical and topological proximity of anterior mesiotemporal components to default and fronto-parietal systems, while posterior hippocampal and parahippocampal regions are more closely implicated in sensory-spatial interactions with external contexts. Spatial gradients that reflect sensory-transmodal differentiation are also evident in subcortical structures (*Müller et al., 2020*; *Tian et al., 2020*) and the cerebellum (*Guell et al., 2018*). Additionally, the cerebellum exhibits a second functional gradient related to multiple-demand processing (*Guell et al., 2018*). Together, our findings add to accumulating evidence that these axes represent core organisational principles of the nervous system (*Nieuwenhuys and Puelles, 2016*). The repetition of gradients at various scales, from the whole cortex to within a brain region, may relate to interconnectivity or result from shared participation in large-scale functional modes.

While we could replicate in-vivo functional imaging findings across two independent datasets, among them the HCP benchmark repository, the singular nature of the BigBrain currently prohibits replication of our 3D histological findings. Follow up work based on ultra-high field magnetic resonance imaging (*DeKraker et al., 2018*; *Yushkevich et al., 2006*) may take similar approaches to explore MTL transitions with respect to in-vivo cortical myeloarchitecture or intrinsic signal flow across cortical depths (*Paquola et al., 2019b*; *Polimeni et al., 2010*; *Huber et al., 2017*; *Finn et al., 2019*), thus allowing assessment of how inter-individual differences in structural and functional organisation of the MTL relate to inter-individual differences in cognitive and affective phenotypes. In addition, this would allow replication in the left hemisphere, which exhibits nuanced cytoarchitectural differences to the right MTL (*Zaidel, 1999*; *Barrera et al., 2001*) and asymmetry in task activations (*Shipton et al., 2014*; *Miller et al., 2018*). In the present work, we tried to optimise alignment to in-vivo fMRI using nonlinear registration and evaluated the accuracy against with individual functional anatomy (*Figure 2—figure supplement 1*). Given that structural and functional perturbations of the MTL are at the core of multiple brain diseases, notably Alzheimer's disease, schizophrenia, and drug-resistant temporal lobe epilepsy (*Braak and Braak, 1991*; *Engel, 2001*; *Kim et al., 2015*; *Lisman et al., 2017*; *Bernhardt et al., 2016*; *Blümcke et al., 2013*), our framework may furthermore advance the understanding and management of these common and severe conditions. By making our MTL confluence model openly available (https://github.com/MICA-MNI/micaopen; *Paquola, 2020*; copy archived at swh:1:rev:2a680f438aa74bd3861e4348a934d6b29cf5bb38) we hope to facilitate such future investigations.

To conclude, our work shows that cytoarchitectural differentiation relates to patterns of intrinsic signal flow within the MTL microcircuit, while the long-axis allows for broad distribution of the computed neural code across the cortical hierarchy. By focusing on one of the most intriguing and integrative regions of the brain, this work illustrates how cytoarchitectural variation can partly account for both the convergence of distributed processing streams and distribution of integrated neural computations across large-scale systems. More broadly, our study of the MTL may provide clues for fundamental aspects of structure-function coupling in the brain (*Mesulam, 1998*; *Lavenex and Amaral, 2000*) and in revealing core principles underlying macroscale cortical organisation (*Sanides, 1962*; *Dart, 1934*; *Abbie, 1938*; *Abbie, 1942*; *Goulas et al., 2019*).

## Materials and methods

### Histological model of cortical confluence

An ultra-high-resolution Merker stained 3D volumetric histological reconstruction of a *post mortem* human brain from a 65-year-old male was obtained from the open-access BigBrain repository, along with pial and white matter surface reconstructions (*Amunts et al., 2013*; *Figure 5A*). We also obtained manually segmented hippocampal subregions (*Figure 5A*), which were labelled with a three-way internal coordinate system; anterior-posterior, proximal-distal and inner-outer based on the solving Laplace's equation (https://osf.io/x542s/) (*DeKraker et al., 2019*). The hippocampal model includes the subiculum and CA1-4. The dentate gyrus was excluded because it is topologically disconnected in the unfolded space (*DeKraker et al., 2019*). The inner surface of the hippocampus, with respect to the hippocampal fissure, is continuous with the pial surface, whereas the outer surface of the hippocampus is continuous with the white matter boundary (*Amaral and Witter, 1989*). Our continuous surface model contains coordinates that represent vertex positions in 3D space and triangles that dictate the connections of vertices into a mesh. We initialised the continuous cortical surface model with inner hippocampal vertices (minimum value on inner-outer axis) and mesiotemporal pial vertices (entorhinal, parahippocampal or fusiform cortex), as well as the triangles that defined their connections in the existing surface meshes. Within this model, we then identified hippocampal bridgehead vertices (minimum value on proximal-distal axis, that is the medial aspect of the subiculum) and matched each to the closest isocortical vertex that had a lower z-coordinate. We only selected inferior vertices to discount the section of the isocortical model that skips over the hippocampus. Then, we created new triangles in the surface mesh that included a hippocampal bridgehead, the matched isocortical vertex and a neighbouring bridgehead on either side, thus creating a bridge between the two surface meshes (*Figure 5C*). At this stage, we also removed all

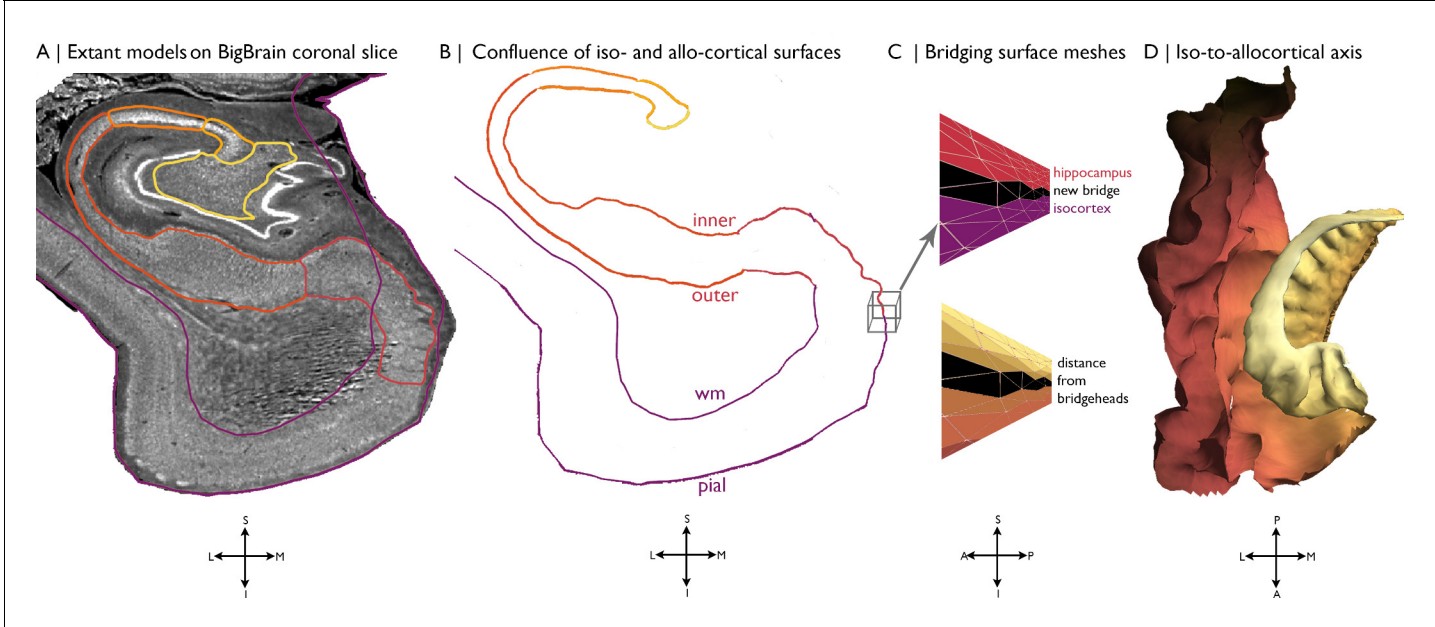

**Figure 5.** Building a continuous cortical surface model of the MTL. (**A**) Isocortical surface models and hippocampal subfield outlines projected on a the 40 μm resolution coronal slice of BigBrain. (**B-C**) We initialised the continuous cortical surface model with inner hippocampal vertices (minimum value on inner-outer axis) and mesiotemporal pial vertices (entorhinal, parahippocampal or fusiform cortex), as well as the triangles that defined their connections in the existing surface meshes. Within this model, we then identified hippocampal bridgehead vertices (medial aspect of the subiculum) and matched each to the closest isocortical vertex that had a lower z-coordinate. We only selected inferior vertices to discount the section of the isocortical model that skips over the hippocampus. Then, we created new triangles in the surface mesh that included a hippocampal bridgehead, the matched isocortical vertex and a neighbouring bridgehead on either side, thus creating a bridge between the two surface constructions. Finally, we calculated the geodesic distance along the continuous surface mesh from each vertex to the nearest hippocampal bridgehead. (**D**) Iso-to-allocortical axis represents the geodesic distance to the hippocampal bridgehead. Orientations are provided below where S = superior, I = inferior, L = lateral, M = medial, A = anterior, p=posterior.

isocortical vertices where the nearest hippocampal bridgehead was inferior, effectively removing the section of the isocortical model that skips over the hippocampus from our continuous surface model. Finally, we calculated the geodesic distance along the continuous surface model from each vertex to the nearest hippocampal bridgehead (*Figure 5C–D*). Negative values represent vertices before the hippocampal bridgehead, and positive values are within the hippocampus. The y-coordinate was used to capture the anterior-posterior axis of the whole MTL. Leveraging the identity of vertices on pial with white and inner with outer surfaces, we then created a continuous model of white and outer surfaces using the same set of triangles. This procedure was developed and evaluated based on visual inspection by a trained anatomist (CP). The entorhinal, parahippocampal, and fusiform areas were defined using the Desikan-Killany atlas, nonlinearly transformed to the BigBrain histological surfaces (*Lewis et al., 2019*), and the hippocampal subfields were assigned in line with manual segmentation (*DeKraker et al., 2019*). Only the right hemisphere was investigated, owing to a tear in the left entorhinal cortex in BigBrain.

## Cytoarchitectural mapping of the cortical confluence

We generated 14 equivolumetric surfaces between the inner and outer confluent surfaces, resulting in 16 surfaces in total. Then, we sampled the intensities from the 40 µm BigBrain blocks along 9432 matched vertices, creating microstructure profiles in the direction of cortical columns for the hippocampus and adjacent MTL areas. The *Merker, 1983* staining is a form of silver impregnation for cell bodies that produces a high contrast of black pigment in cells on a virtually colourless background. Stained histological sections were then digitized at 20 µm, resulting in greyscale images with darker colouring where many or large cells occur. The density and size of cells vary across laminar, as well as areas, thus capturing the regional differentiation of cytoarchitecture. In lieu of precise laminar decomposition, which is currently untenable in such transition regions (*Wagstyl et al., 2020*), we utilised an equivolumetric algorithm to optimise the fit of depth-wise surfaces with variations in thickness and curvature (*Waehnert et al., 2014*). For the sake of clarity, we refer to the surfaces by the percentage depth of their initialisation, which was adjusted for curvature to become equivolumetric.

Unsupervised machine learning tested the hypothesis that the iso-to-allocortical axis is the principle gradient of cytoarchitectural differentiation in the MTL. The procedure involved calculating microstructure profile covariance between all vertices then extracting the principle axis of cytoarchitectural variation as the first eigenvector from diffusion map embedding (*Paquola et al., 2019b*; *Vos de Wael et al., 2020*; *Coifman et al., 2005*). Diffusion map embedding is optimally suited for multi-scale architectures, because unlike global methods, such as principle component analysis and multidimensional scaling, local geometries are preserved and then integrated into a set of global eigenvectors. The statistical relationship between the principle eigenvector and the proximal-distal axis was modelled with a series of increasingly complex polynomial curves. We selected the model at the inflection point of adjusted $R^2$ values, taken across polynomials of 1–5. We extracted the residuals from the selected model to determine which regions deviated from the expected association.

The defining cytoarchitectural features of the iso-to-allocortical axis were established using feature selection in a random forest regression. A set of 20 cytoarchitectural features (16 depth-wise intensities and 4 central moments [*Paquola et al., 2019a*]) were z-standardised and fed into random forest regression to predict iso-to-allocortical axis. Hyperparameters were selected via 5-fold cross validation in a grid search. We trained the model on 70% of vertices then measured prediction accuracy by $R^2$ with the remaining 30%. Feature importance was estimated as the mean decrease in variance, and features with higher than average importance were selected. These feature importance values must be taken in the context of the model, and they represent the relative variance explained within the model. The procedure was repeated across 100 splits to assess robustness of prediction accuracy and feature importance. Using the same hyperparameters, we refit the model with only the selected features and assessed the predictive accuracy in the same manner. For post-hoc inference, we assessed the pattern of change in each selected feature with the iso-to-allocortical axis using a cubic polynomial (the best fit in the unsupervised model; *Figure 1F*).

To assess the consistency of iso-to-allocortical variations along the anterior-posterior axis, we repeated our analyses with stratification by the anterior-posterior axis. We fit a cubic polynomial between each selected feature and the iso-to-allocortical axis within 1 mm coronal slices of

confluence model and approximated the goodness of fit (adjusted $R^2$) for the 23 coronal slices that included the hippocampus proper.

## Registration to MNI152 standard atlas

The cortical confluence model was non-linearly registered to the MNI152 2 mm standard atlas to facilitate targeted associations to in-vivo functional imaging. This procedure involved labelling voxels in native histological space (1000 µm) according to their position along the iso-to-allocortical axis. To allow for downsampling (40→1000 µm), we used average axis values of all vertices contained within a voxel. Next, we applied a series of one linear and two nonlinear transformations to register the volumetric representation of cortical confluence to MNI152 standard atlas. The first two transformations are available via the BigBrain repository (ftp://bigbrain.loris.ca/BigBrainRelease.2015/3D_Volumes/MNI-ICBM152_Space/), and the final nonlinear optimisation is available as part of an open science framework (*Xiao et al., 2019*; https://osf.io/xkqb3/). Finally, we restricted the model to the anterior-posterior range of the hippocampus proper to allow appropriate comparisons of both geometric axes and used a grey matter partial volume mask to ensure the standardised volumetric atlas conformed to cortical boundaries. Notably, only the right MTL was modelled and all subsequent analyses relate to the right MTL.

## Functional MRI acquisition and preprocessing

We studied resting state functional imaging data from 40 healthy adults (microstructure informed connectomics (MICs) cohort; 14 females, mean ± SD age = 30.4 ± 6.7, two left-handed). Participants gave informed consent and the study was approved by the local Research Ethics Board of the Montreal Neurological Institute and Hospital. MRI data was acquired on a 3T Siemens Magnetom Prisma-Fit with a 64-channel head coil. A submillimetric T1-weighted image was acquired using a 3D-MPRAGE sequence (0.8 mm isotropic voxels, 320 × 320 matrix, 24 sagittal slices, TR = 2300 ms, TE = 3.14 ms, TI = 900 ms, flip angle = 9°, iPAT = 2). One 7 min rs-fMRI scan was acquired using multi-band accelerated 2D-BOLD echo-planar imaging (TR = 600 ms, TE = 30 ms, 3 mm isotropic voxels, flip angle = 52°, FOV = 240 × 240 mm$^2$, slice thickness = 3 mm, mb factor = 6, echo spacing = 0.54 mms). Participants were instructed to keep their eyes open, look at fixation cross, and not fall asleep. All fMRI data underwent gradient unwarping, motion correction, distortion correction, brain-boundary-based registration to structural T1-weighted scan and grand-mean intensity normalization. The rs-fMRI data was additionally denoised using an in-house trained ICA-FIX classifier (*Salimi-Khorshidi et al., 2014*; *Griffanti et al., 2014*), which notably outperforms motion scrubbing and spike regression in reproducing resting state networks and conservation of temporal degrees of freedom (*Pruim et al., 2015a*; *Pruim et al., 2015b*). Timeseries were sampled on native cortical surfaces and averaged within 1000 spatially contiguous, functionally defined parcels (*Schaefer et al., 2018*). Finally, for each subject, we generated a nonlinear transformation (MNI152 to native rs-fMRI space) (*Andersson et al., 2007*) and a binary grey matter mask (>50% tissue-probability in T1w image) to facilitate mapping the standardised volumetric atlas to individual subjects' functional images.

## Dynamic modelling

Spectral dynamic causal modelling estimated the effective connectivity between regions of the cortical confluence model (*Friston et al., 2014*; *Razi et al., 2015*). This approach involves a generative model with one state depicting the interactions of neuronal populations and another state providing a conventional linear convolution of neuronal activity to the hemodynamic response (*Stephan et al., 2007*). We specified a fully connected dynamic model, without exogenous inputs, and estimated model parameters, that is the effective connectivity between regions, and their marginal likelihood using a Variational Laplace inversion (*Friston et al., 2007*). In line with previous research (*Friston et al., 2014*; *Razi et al., 2015*), we used a fourth order autoregressive model to estimate cross spectra of the timeseries. The model was built for each of the 40 subjects separately, and group-level estimates were calculated using a second-level parametric empirical Bayesian model (*Friston et al., 2016*). The standardised volumetric atlas of the iso-to-allocortical axis was divided into a set of discrete bins and transformed to individual functional space, then BOLD timeseries were extracted and the mean timeseries was calculated for each bin. We evaluated the optimal

number of bins, ranging from 4 and 14, and selected the finest resolution that still yielded a hierarchical model for which inversion was computationally feasible using a second-level parametric empirical Bayesian analysis. The percentage of voxels of each bin that occupied anatomical subregions was calculated based on the Desikan-Killany atlas and hippocampal subfield segmentations in Free-Surfer (*Iglesias et al., 2015*; *Desikan et al., 2006*). Next, we applied Bayesian model reduction to the posterior densities over the second-level parameters, involving a greedy search over all permutations of parameter inclusion, to discover the structure of the optimal sparse graph.

To test the hypothesis that the iso-to-allocortical axis constrains the signal flow through the mesiotemporal confluence, we compared the strength of effectivity connectivity with the concordance of an edge to the iso-to-allocortical axis using product-moment correlation coefficient. Each edge was labelled by the degree of deviation from the iso-to-allocortical axis (0–6 in the 8-bin model). We further stratified these edges by the direction along the axis (*i.e.*, signal flow towards isocortex or towards allocortex) and compared the correlation coefficients in each direction (*Meng et al., 1992*). Next, we divided the standardised volumetric atlas into thirds along the anterior-posterior axis and repeated the dynamic model with the preselected number of bins for each third separately. The tripartite division was chosen to roughly approximate the head, body and tail division of the hippocampus (*Iglesias et al., 2015*; *Van Leemput et al., 2009*), for consistency with previous work on hippocampal connectivity (*Vos de Wael et al., 2018*; *Libby et al., 2012*), and to strike a balance between a fine-grained approach with comparability to the main model. The tripartite division captured similar subfield arrangement to the whole model (*Figure 2—figure supplement 1A*). We assessed consistency across the anterior-posterior axis by calculating the product-moment correlation coefficient of the effective connectivity estimates from non-reduced models between y-axis restricted and the full mesiotemporal confluence model, as well as repeating the association of the effective connectivity with deviation from the iso-to-allocortical axis in the reduced model obtained from Bayesian model reduction. Additionally, to assess the mapping of the iso-to-allocortical axis to individual fMRI scans, we calculated the functional homogeneity of rs-fMRI timeseries within bins in native subject space then averaged across subjects (*Figure 2—figure supplement 1B*). Across individuals, we were able to show higher functional homogeneity within bins than between, in line with the range of a previous study using individualised structural segmentation (*Zhong et al., 2019*), supporting the accurate mapping of the axis to individual anatomy.

## Extrinsic functional connectivity

The association between iso-to-allocortical axis and macroscale rs-fMRI connectivity patterns was estimated via a product-moment correlation for each isocortical parcel. Positive r values reflect higher connectivity towards the allocortical end, whereas negative r values reflect higher connectivity towards the isocortical end. We compared the spatial pattern of r values across the isocortex with canonical functional gradients (*Margulies et al., 2016*). To this end, we calculated parcel-parcel rs-fMRI connectivity matrix for each participant, averaged across the cohort, transformed this into a normalised angle matrix, and subjected it to diffusion map embedding. The principle eigenvectors, referred to as functional gradients, resembled previous descriptions (*Margulies et al., 2016*; *Paquola et al., 2019b*; *Karapanagiotidis, 2019*). Spearman correlations assessed the spatial correspondence of the r map with the functional gradients and spin permutations established significance, while accounting for spatial autocorrelation (*Váša et al., 2018*; *Alexander-Bloch et al., 2018*). We tested the specificity of the associations with functional gradients by comparing the correlation coefficients (*Meng et al., 1992*). We repeated this comparison within thirds of the anterior-posterior axis and tested the association of rs-fMRI connectivity to the isocortex with the anterior-posterior axis.

## Effects of SNR

We assessed potential effects of SNR variations on macroscale MTL connectivity. Here, spatial SNR was taken as the mean signal within a voxel divided by the standard deviation of the signal across all grey matter voxels. Temporal SNR was taken as the mean signal of a voxel divided by the standard deviation of signal in that voxel over time. Additionally, we assessed the contribution of the temporal lobe to the relationship between spatial maps and functional gradients by excluding the lobe for computing the final correlation. We used the Deskian-Killany atlas to define the temporal lobe and excluded isocortical parcel with any vertices in the temporal lobe (n = 196).

## Replication

We replicated the fMRI findings in 40 unrelated healthy adults (14 females, age = 29.4 ± 3.8 years) from the minimally preprocessed S900 release of the Human Connectome Project (HCP) (*Glasser et al., 2013*). Specifically, we tested whether effective connectivity was associated with the deviation from the iso-to-allocortical axis, in the whole hippocampus and within coronal slices, whether rs-fMRI along the iso-to-allocortical axis was spatially correlated with the multiple-demand functional gradient and whether rs-fMRI along the anterior-posterior axis was spatially correlated with the sensory-transmodal functional gradient.

## MRI acquisition

In brief, MRI data were acquired on the HCP's custom 3T Siemens Skyra equipped with a 32-channel head coil. Two T1-weighted images with identical parameters were acquired using a 3D-MPRAGE sequence (0.7 mm isotropic voxels, matrix = 320 × 320, 256 sagittal slices; TR = 2,400 ms, TE = 2.14 ms, TI = 1,000 ms, flip angle = 8°; iPAT = 2). Four rs-fMRI scans were acquired using multiband accelerated 2D-BOLD echo-planar imaging (2 mm isotropic voxels, matrix = 104 × 90, 72 sagittal slices; TR = 720 ms, TE = 33 ms, flip angle = 52°; mb factor = 8; 1200 volumes/scan). We utilised the first session of rs-fMRI. Participants were instructed to keep their eyes open, look at fixation cross, and not fall asleep. For rs-fMRI, the timeseries were corrected for gradient nonlinearity and head motion.

## MRI processing

The R-L/L-R blipped scan pairs were used to correct for geometric distortions. Distortion-corrected images were warped to T1w space using a combination of rigid body and boundary-based linear registrations (*Greve and Fischl, 2009*). These transformations were concatenated with the transformation from native T1w to MNI152 to warp functional images to MNI152. Further processing removed the bias field (as calculated for the structural image), extracted the brain, and normalised whole-brain intensity. A high-pass filter (>2,000 s FWHM) corrected the timeseries for scanner drifts, and additional noise was removed using ICA-FIX (*Salimi-Khorshidi et al., 2014*). Tissue-specific signal regression was not performed (*Murphy and Fox, 2017*). Using the functional images warped to MNI152 space, timeseries were extracted from all voxels encompassed by the cortical confluence atlas and 1000 isocortical parcels of the Schaefer 7-network atlas (*Schaefer et al., 2018*). Timeseries were averaged within each isocortical parcel.

## fMRI analysis

To analyze intrinsic circuitry, we tested whether effective connectivity was stronger on the iso-to-allocortical axis, using an 8bin iso-to-allocortical division of the entire MTL. We repeated this analysis within three consecutive segments of the long axis. To analyze macroscale connectivity, we calculated product-moment correlation coefficients between connectivity patterns of specific iso-to-allocortical MTL subsections and functional gradients. The isocortical functional gradients were regenerated within the HCP cohort and were highly similar to the original dataset (Spearman correlation between datasets: $r_{G1} = 0.78$, $p_{spin} < 0.001$, $r_{G3} = 0.81$, $p_{spin} < 0.001$).

## Ethical Statement

Participants whose data was studied in the main analysis provided informed consent, and the study was approved by the Research Ethics Board of the Montreal Neurological Institute and Hospital (2018–3469). Participants from the HCP dataset provided informed consent for open sharing of their deidentified data, approved by the Washington University Institutional Review Board as part of the HCP.

## Acknowledgements

Casey Paquola was funded through a postdoctoral fellowship of the Fonds de la Recherche due Quebec – Santé (FRQ-S). Oualid Benkarim was funded by a Healthy Brains for Healthy Lives (HBHL) postdoctoral fellowship. Boris Bernhardt acknowledges research support from the National Science and Engineering Research Council of Canada (NSERC Discovery-1304413), the Canadian Institutes of Health Research (CIHR FDN-154298), SickKids Foundation (NI17-039), Azrieli Center for Autism

Research (ACAR-TACC), FRQ-S, and the Tier-2 Canada Research Chairs program. Andrea Bernasconi and Neda Bernasconi were funded by CIHR and NSERC. Jessica Royer was funded by a CIHR fellowship. Jonathan Smallwood was funded by the European Research Council (WANDERING-MINDS). Sara Lariviere acknowledges funding from Fonds de la Recherche du Québec – Santé (FRQ-S) and the Canadian Institutes of Health Research (CIHR). We would also like to acknowledge support from the Helmholtz Foundation and the Healthy Brains for Healthy Lives initiative. Data were provided, in part, by the Human Connectome Project, WU-Minn Consortium (Principal Investigators: David Van Essen and Kamil Ugurbil; 1U54MH091657) funded by the 16 NIH Institutes and Centers that support the NIH Blueprint for Neuroscience Research; and by the McDonnell Center for Systems Neuroscience at Washington University. Alan Evans and Boris Bernhardt were supported by the Helmholtz International BigBrain Analytics Learning Laboratory (HIBALL), funded by the Helmholtz Foundation and Healthy Brains and Healthy Lives (HBHL).

## Additional information

### Funding

| Funder | Grant reference number | Author |
| --- | --- | --- |
| Canadian Institutes of Health Research | CIHR FDN-154298 | Boris C Bernhardt |
| Fonds de Recherche du Québec - Santé | postdoctoral fellowship | Casey Paquola |
| Healthy Brains for Healthy Lives | postdoctoral fellowship | Oualid Benkarim |
| NSERC | NSERC Discovery-1304413 | Boris C Bernhardt |
| Sick Kids Foundation | NI17-039 | Boris C Bernhardt |
| Azrieli Center for AutismResearch | ACAR-TACC | Boris C Bernhardt |
| Canada Research Chairs | Tier-2 | Boris C Bernhardt |
| CIHR | | Andrea Bernasconi Neda Bernasconi |
| NSERC | | Andrea Bernasconi Neda Bernasconi |
| CIHR | Fellowship | Jessica Royer |
| European Research Council | WANDERINGMINDS | Jonathan Smallwood |
| Fonds de la Recherche du Québec - Santé | Acknowledges funding | Sara Larivière |
| CIHR | Acknowledges funding | Sara Larivière |
| Helmholtz Association & Healthy Brains for Healthy Lives | Helmholtz International BigBrain Analytics Learning Laboratory (Hiball) | Alan C Evans Boris C Bernhardt |
| Australian Research Council | DE170100128 | Adeel Razi |
| Australian Research Council | DP200100757 | Adeel Razi |
| National Health and Medical Research Council | Investigator Grant 1194910 | Adeel Razi |

The funders had no role in study design, data collection and interpretation, or the decision to submit the work for publication.

### Author contributions

Casey Paquola, Conceptualization, Software, Formal analysis, Investigation, Visualization, Methodology, Writing - original draft, Writing - review and editing; Oualid Benkarim, Adeel Razi, Formal analysis, Writing - review and editing; Jordan DeKraker, Jessica Royer, Shahin Tavakol, Ali Khan, Data curation, Writing - review and editing; Sara Larivière, Stefan Frässle, Sofie Valk, Andrea Bernasconi,

Neda Bernasconi, Alan C Evans, Writing - review and editing; Jonathan Smallwood, Conceptualization, Writing - review and editing; Boris C Bernhardt, Conceptualization, Supervision, Funding acquisition, Writing - original draft, Project administration, Writing - review and editing

### Author ORCIDs
Casey Paquola (iD) https://orcid.org/0000-0002-0190-4103
Sara Larivière (iD) https://orcid.org/0000-0001-5701-1307
Stefan Frässle (iD) http://orcid.org/0000-0002-8011-2226
Adeel Razi (iD) http://orcid.org/0000-0002-0779-9439
Boris C Bernhardt (iD) https://orcid.org/0000-0002-9536-7862

### Ethics
Human subjects: Participants gave informed consent and the study was approved by the local Research Ethics Board of the Montreal Neurological Institute and Hospital (2018-3469).

### Decision letter and Author response
Decision letter https://doi.org/10.7554/eLife.60673.sa1
Author response https://doi.org/10.7554/eLife.60673.sa2

## Additional files

### Supplementary files
• Transparent reporting form

### Data availability
Code and data related to this specific project are openly available under https://github.com/MICA-MNI/micaopen/tree/master/cortical_confluence (copy archived at https://archive.softwareheritage.org/swh:1:rev:2a680f438aa74bd3861e4348a934d6b29cf5bb38/), BigBrain related information are openly available under https://bigbrain.loris.ca/main.php. The human connectome project dataset is available under https://db.humanconnectome.org/.

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
