## [Decision Letter]

**Acceptance summary:**

The paper describes an exciting and innovative method of bridging post-mortem cytoarchitecture with in vivo functional MRI, allowing for a powerful and compelling investigation of micro-architecture in the medial temporal lobe (MTL). This work has important implications for how information transfer occurs through macro-structural circuits across the whole brain as well as more local brain circuits in the MTL.

**Decision letter after peer review:**

Thank you for submitting your article "Convergence of cortical types and functional motifs in the mesiotemporal lobe" for consideration by *eLife*. Your article has been reviewed by three peer reviewers, and the evaluation has been overseen by a Reviewing Editor and Timothy Behrens as the Senior Editor. The following individuals involved in review of your submission have agreed to reveal their identity: James M Shine (Reviewer #1); Rosanna Olsen (Reviewer #2); Edward Hogan (Reviewer #3).

The reviewers have discussed the reviews with one another and the Reviewing Editor has drafted this decision to help you prepare a revised submission.

All three reviewers saw great merit in your work and were enthusiastic about its potential. Nonetheless, each reviewer raised several substantive concerns, which much be addressed in a revision. The individual reviews are provided below. Broadly speaking, we see the essential revision as (1) providing additional clarity with respect to methods, (2) further unpacking of some of the results, as well as conducting a few targeted statistical analyses (i.e., to test for differences in slopes), and (3) clearer positioning of the current work as it relates to the existing literature. We hope you find these reviews helpful as you revise the manuscript, and we look forward to reading the revision.

Reviewer #1:

Thank you for inviting me to review this manuscript by Paquola and colleagues, in which the authors used a combination of high-resolution anatomical data, machine learning, spectral DCM and resting functional connectivity measures to interrogate the relationship between structural and functional gradients of organization within the mesial temporal lobe.

The study is broken into four related sections. In the first section, the authors analysed vertices within a set of mesial temporal lobe structures using a random-forest algorithm, which identified a set of microstructural profiles across the structure. They then interrogated these profiles for evidence of an iso-to-allometric axis, which is a principle known to characterise the transition from 6-layered isocortex (in entorhinal cortex) to 3-layer allocortex (in the hippocampal formation). The authors found evidence consistent with this transition in the BigBrain data, particularly with respect to the skewness of the distribution of thickness across the layers.

In the second section, the authors use Spectral DCM on resting state data from a group of 40 individuals. They then relate the results of the spectral DCM model to the gradients identified using structural anatomy. This section was well-motivated and conducted.

In the third section, the authors compare the structural gradient to resting state functional connectivity with vertices within the cerebral cortex. The results here were quite compelling, showing a dissociation between the iso- and allo-cortical poles in the MTL in which the iso-cortex was correlated with fluctuations in the lateral dorsal attention and frontoparietal networks, whereas the allo-cortical pole was correlated with vertices in the default mode and medial occipital regions.

In the final section, the authors conducted a number of checks of their analysis, including an SNR test to ensure that the temporal lobes (a notorious site for MRI signal dropout) were adequate, and a substantial replication analysis. They should be commended for these steps, and also for making their code freely available.

Comments

1) Section 1: I wonder whether the manuscript might benefit from the unpacking of the random forest results. Is there an intuitive way to characterize skewness that may benefit the reader – such as a particularly uneven spread of thickness distributed across the layers? And is this finding something that we might expect, given the hypothesized gradient of iso-to-allocortex in the MTL?

2) Section 1: Along these lines, is it fair to single out an individual measure from the random-forest regression as being the most salient? From my understanding (which might be mistaken), the weights on a particular variable in a regression need to be viewed in context of the performance of the whole model.

3) Section 2: One minor comment is that it might helpful for the reader if the "in" and "out" effective connectivity directions were incorporated into the matrix in Figure 2A.

4) Section 2: I wasn't sure that I followed the logic of the experiment in which the authors split the MTL data into thirds to test for the consistency of their results. Were each of these sufficiently powered to allow for direct comparison with the main effect? Did the boundaries between these models cut across known regional areas? Perhaps a different way to achieve the same ends would be to use bootstrapping in order to provide a confidence interval around the relationship between structure and function?

5) Section 3: Did the authors hypothesize the iso vs. allo-cortical relationship to resting state networks a priori, or was it discovered upon exploration of the data. Either is fine, in my opinion, but I think it would benefit the reader to have these results placed in the context of the known literature.

6) Section 3: Do the authors expect that the patterns identified in the MTL will relate to subcortical gradients identified in other structures, such as the cerebellum (Guell et al., 2018), thalamus (Müller et al., 2020, and basal ganglia (Stanley et al., 2019)? See also Tian et al., 2020 for general subcortical gradients.

Reviewer #2:

This paper does a very good job of underscoring the importance of characterizing the structural organization of the cortex at a deep level in order to inform functional organization. The authors present an exciting and innovative method of bridging post-mortem cytoarchitecture with in vivo functional MRI, allowing for a powerful and compelling investigation of MTL micro-architecture. This work has important implications for how information transfer occurs through macro-structural and more local brain circuits. The two major findings regarding the allo-iso and the anterior-posterior gradient are supported by the previous literature, but so far characterization of this organization in humans in vivo has been somewhat limited. Most of my suggestions below are regarding points that could be clarified or methods that were unclear.

1) Was there an a-priori prediction regarding the "multi-demand" network? This part of the narrative seemed to come out of the blue and could use more background.

2) Some of the methods are not fully described and are hard to understand. For example, the surface models that are used to sample and model the properties of the microstructure at different cortical depths could be described in more detail. I was also having trouble understanding two things about the "confluence" or "intersection" between the allocentric and isocentric cortices. I was left wondering if the intersection is defined as a plane in surface space, demarcating the separation between hippocampus and entorhinal cortex? Is the confluence/intersection defined based on the manual hippocampal subfields (i.e. medial boundary of the subiculum) or is it defined some other way using the surface profiles/features? Finally, how is geodesic "distance" computed? I would suggest adding a figure to give an overview of these aspects of the methods.

3) Related to the point above, I get the impression that these data show there is no strict boundary between the allo and iso-cortex but rather that there is a somewhat smooth gradient. This point could be made more clear in the Abstract and Discussion. What implications does this particular finding have for theories of MTL subregion function?

4) When r-values are reported to differ for different gradients (e.g. iso versus allo) it is important to test for a significant difference in the slopes (e.g. Fisher r-to-z transform or similar) to know if the relationships are statistically different from one another.

5) This paper builds nicely on other work by DeKraker and colleagues (2019) that has analyzed the microstructural properties of the hippocampus. I think the readers of this paper would appreciate a brief description of how this investigation is similar/different from that work. For example, are the "features" identified here largely overlapping with those identified by DeKraker, and if not, how do they differ here?

6) In the effective connectivity analysis of the MTL, how is variability of the MTL anatomy taken into account? For example, the fusiform and parahippocampal regions of interest will contain highly variable anatomical structures across subjects (e.g. different folding patterns of the collateral sulcus). Given that the focus on anatomical specificity is a major strength of this paper, I would be curious to know how anatomical variability/specificity is accounted for when the data are morphed into MNI152 volume space.

7) I was unsure which analyses were replicated in the Human Connectome Project (HCP) dataset. It is stated that the isocortical functional gradients were re-generated within the HCP cohort and that results were "highly similar" to the original dataset. Was this similarity formally tested?

Reviewer #3:

General assessment of the work: The authors report a study of the mesial temporal lobe (MTL), particularly focusing on structural/functional changes related to transition regions from six layer isocortex to three layered allo-cortex. This group uses their expertise in imaging processing techniques to define the anatomical regions of the mesial temporal lobe transition from isocortex to allocortex using the BigBrain high-resolution histological reconstruction. Using this single high-resolution histological image, they show intensity changes which correlate with the isocortex/allocortex transition. They then use this high resolution reconstruction to coregister to rs-fMRI, and define effective connectivity within the mesiotemporal lobe. Finally, they show variation rs-fMRI global patterns in relationship to the iso-to-allocortical axis, as well as the mesial temporal a/p axis.

Substantive concerns:

This is an interesting study which shows novel relationships between mesial temporal structures and whole brain functional organization. As the authors point out, the novel part of the study involves defining cytoarchitectural regions, and correlating these changes with both local and global function as defined by BOLD fMRI. This is a novel study examining the iso-allocortical transitions with the MTL, and correlating them with local and global rs-fMRI changes. As the authors state, the global rs-fMRI findings related the anterior-posterior axis of the MTL are not new, but add complimentary findings in comparison the iso-allocortical transition findings. Given this, I will focus my comments on the use of the BigBrain image, and definition of the MTL transitions for use in defining regions in the rs-fMRI images.

1) With the BigBrain data, only the right hippocampus was used for segmentation, due to a rip in the histopathological sections of entorhinal cortex on the left. It is therefore assumed that the right MTL segmentations were inverted and also used for the left MTL rs-fMRI analysis. If this is the case, it should be more clearly stated in the Materials and methods. Also, discussion should be added to the possible implications for results, both in respect to replicating the histological intensity findings (which could be tested in two hippocampi if both right and left were processed) and the known structural differences between the right and left hippocampi.

2) I had concerns that using the higher resolution BigBrain image as a template for the 8 nodes in the MTL for the much lower resolution rs-fMRI images would be problematic for signal to noise ratio. However, the authors have convincingly shown consistent findings when controlling for signal to noise ratios.

3) The authors mention (and reference) the correlation of histopathological cellular staining intensities with cellular densities and soma size in the Materials and methods section. Given the centrality of this concept to their findings of the BigBrain data, some addition to the discussion about this concept and the underlying evidence for correlation of staining intensity and cellular densities and soma size would be helpful.

---

## [Author Response]

All three reviewers saw great merit in your work and were enthusiastic about its potential. Nonetheless, each reviewer raised several substantive concerns, which much be addressed in a revision. The individual reviews are provided below. Broadly speaking, we see the essential revision as (1) providing additional clarity with respect to methods, (2) further unpacking of some of the results, as well as conducting a few targeted statistical analyses (i.e., to test for differences in slopes), and (3) clearer positioning of the current work as it relates to the existing literature. We hope you find these reviews helpful as you revise the manuscript, and we look forward to reading the revision.

We would like to thank the Editor and reviewers for their positive evaluation and helpful suggestions. In the updated manuscript, we have (1) expanded and further clarified the Materials and methods, including the addition of a new figure explaining the construction of the continuous surface model, (2) further unpacked the results and performed statistical tests on the difference between slopes, and (3) expanded our Discussion of the findings in relation to previous literature on hippocampal subfields, subcortical gradients, the BigBrain hippocampus (DeKraker et al., 2019) and hemispheric asymmetry of the MTL. We have replied to each comment in turn below and highlighted corresponding passages in yellow in the revised manuscript.

Reviewer #1:Comments1) Section 1: I wonder whether the manuscript might benefit from the unpacking of the random forest results. Is there an intuitive way to characterize skewness that may benefit the reader – such as a particularly uneven spread of thickness distributed across the layers? And is this finding something that we might expect, given the hypothesized gradient of iso-to-allocortex in the MTL?

We thank the reviewer for the comment. The skewness feature pertains to the relative differences in staining intensity across cortical depths, and as such is sensitive to depth-dependent microstructural variations. By expanding Figure 1—figure supplement 2, we were able to further illustrate which depth-dependent differences relate to skewness within the MTL, specifically. We found that low skewness (pink) reflects a flat profile with intensity dip in the upper surfaces, midrange skewness (mustard) reflects a profile with a mid-depth peak, and high skewness (emerald) reflects a flat profile with an intensity peak in the upper surfaces. Thus, we surmise that the skewness is sensitive to profile flatness and the relative intensity of upper surfaces within the MTL. Indeed, we had hypothesised that hippocampal allocortex would exhibit a flatter profile than isocortex in the MTL, related to the low laminar differentiation of the allocortex. We have updated Figure 1—figure supplement 2, Introduction, Results, and Discussion with these details:

“During ontogeny, isocortex (from iso- "equal") differs from allocortical (from allo- "other") hippocampus, which lacks clear lamination prenatally and exhibits a more even distribution of neurons across depths”

“Microstructure profile skewness increased from iso-to-allocortex, which pertained to a shift from a profile with mid-depth peak to a flatter microstructure profile (Figure 1H, see Figure 1—figure supplement 2 for exemplar microstructure profiles and feature values).”

“Here, a key feature to predict the iso-to-allocortical transition in our model based on microstructure profiling was skewness. Profile skewness, reflecting relative staining intensity across cortical depths, captured the expected shift from a peaked isocortical profile to a flatter allocortical microstructure profile.”

2) Section 1: Along these lines, is it fair to single out an individual measure from the random-forest regression as being the most salient? From my understanding (which might be mistaken), the weights on a particular variable in a regression need to be viewed in context of the performance of the whole model.

We agree with the reviewer that feature salience is contextualized in the overall model, and have further moderated our interpretation of the feature importance by describing skewness as “a key feature to predict iso-to-allocortical transition in our model ” in the updated Discussion (see previous reply). We have also expanded upon the rationale and validity for this approach in the Materials and methods:

“Feature importance was estimated as the mean decrease in variance, and features with higher than average importance were selected. These feature importance values must be taken in the context of the model, and they represent the relative variance explained within the model.”

3) Section 2: One minor comment is that it might helpful for the reader if the "in" and "out" effective connectivity directions were incorporated into the matrix in Figure 2A.

We’ve added the seed and target labels to Figure 2 as recommended.

4) Section 2: I wasn't sure that I followed the logic of the experiment in which the authors split the MTL data into thirds to test for the consistency of their results. Were each of these sufficiently powered to allow for direct comparison with the main effect? Did the boundaries between these models cut across known regional areas? Perhaps a different way to achieve the same ends would be to use bootstrapping in order to provide a confidence interval around the relationship between structure and function?

We opted for a tripartite division of the MTL, to approximate the head, body and tail division of the hippocampus (Van Leemput et al., 2009; Iglesias et al., 2015), for consistency with previous work on hippocampal connectivity (Libby et al., 2012; Vos de Wael et al., 2018), and to strike a balance between a fine-grained approach with comparability to the main model. The tripartite division captured similar subfield arrangement to the whole model (see updated Figure 2—figure supplement 1), although the entorhinal cortex was not captured by the posterior third. Thus, further subdivision would result in greater mismatch of regional composition between the tripartite and whole models. Of note, we derived the DCM from average timeseries of each bin, and as such the AP split holds the same power as the full analysis. There is approximately one third of the number of voxels that contribute to the average timeseries, though.

We’ve expanded upon this decision in the Materials and methods:

“Next, we divided the standardised volumetric atlas into thirds along the anterior-posterior axis and repeated the dynamic model with the preselected number of bins for each third separately. The tripartite division was chosen to roughly approximate the head, body and tail division of the hippocampus (Van Leemput et al., 2009; Iglesias et al., 2015), for consistency with previous work on hippocampal connectivity (Libby et al., 2012; Vos de Wael et al., 2018), and to strike a balance between a fine-grained approach with comparability to the main model. The tripartite division captured similar subfield arrangement to the whole model (Figure 2—figure supplement 1A)”

5) Section 3: Did the authors hypothesize the iso vs. allo-cortical relationship to resting state networks a priori, or was it discovered upon exploration of the data. Either is fine, in my opinion, but I think it would benefit the reader to have these results placed in the context of the known literature.

MTL is increasingly recognised to extend beyond memory, we hypothesised that the isoto-allocortical axis would capture global features of cortical function that are related to different types of cognition and exhibit distinct functional connectivity to the anterior-posterior axis. We did not, however, have a clear hypothesis on the specific manner in which large-scale networks would be represented in the iso-to-allocortical axis. We’ve clarified this in the Introduction:

“In this way, the coarse representations in the MTL could be mirrored by neural motifs present across the broader cortical system, helping to explain the regions’ contribution to multiple aspects of memory (Clark, 2018), and to cognitive maps describing the structure of many different tasks (Eichenbaum and Cohen, 2014; Tavares et al., 2015; Solomon et al., 2019). […] Extensive research on preferential connectivity of anterior hippocampus to transmodal isocortex and posterior hippocampus to sensory isocortex shows that the MTL architecture can reflect a large-scale functional motif (Libby et al., 2012; Strange et al., 2014; Maass et al., 2015; Vos de Wael et al., 2018; Przeździk et al., 2019). Our study explored whether the iso-to-allocortical axis also reflects a large-scale functional motif, and whether the axis interactions could help understand how the MTL elicits a broad role in neural function.”

6) Section 3: Do the authors expect that the patterns identified in the MTL will relate to subcortical gradients identified in other structures, such as the cerebellum (Guell et al., 2018), thalamus (Müller et al., 2020, and basal ganglia (Stanley et al., 2019)? See also Tian et al., 2020 for general subcortical gradients.

We thank the reviewer for pointing us towards this interesting literature. Across prior studies in different subcortical structures, several have suggested an existence of a principle spatial gradient that also reflects sensory-transmodal differentiation (Guell et al., 2018; Müller et al., 2020), which aligns with the anteriorposterior axis of the MTL. Additionally, in the cerebellum a second gradient differentiates networks involved in task focused and unfocused processing (Guell et al., 2018), similar to the multiple demand gradient that we found to co-vary with the iso-to-allocortical axis. We’ve expanded upon this in the revised Discussion.

“Sensory-transmodal differentiation of rs-fMRI connectivity along the MTL long axis is well-evidenced in the literature (Libby et al., 2012; Strange et al., 2014; Maass et al., 2015; Vos de Wael et al., 2018; Przeździk et al., 2019) and indicates more anatomical and topological proximity of anterior mesiotemporal components to default and frontoparietal systems, while posterior hippocampal and parahippocampal regions are more closely implicated in sensoryspatial interactions with external contexts. Spatial gradients that reflect sensory-transmodal differentiation are also evident in subcortical structures (Müller et al., 2020; Tian et al., 2020) and the cerebellum (Guell et al., 2018). Additionally, the cerebellum exhibits a second functional gradient related to multiple demand processing, similar to the iso-to-allocortex axis of the MTL (Guell et al., 2018). Together, our findings add to accumulating evidence that these axes represent core organisational principles of the nervous system (Nieuwenhuys and Puelles, 2016). The repetition of gradients at various scales, from the whole cortex to within a brain region, may relate to interconnectivity or result from shared participation in large-scale functional modes.”

Reviewer #2:1) Was there an a-priori prediction regarding the "multi-demand" network? This part of the narrative seemed to come out of the blue and could use more background.

We had not *a priori* predicted that the iso-to-allocortical axis would reflect the multi-demand gradient, however, we were motivated by findings on the anterior-posterior axis to explore the associations with large-scale functional gradients. We’ve clarified this in the Introduction:

“In this way, the coarse representations in the MTL could be mirrored by neural motifs present across the broader cortical system, helping to explain the regions’ contribution to multiple aspects of memory (Clark, 2018), and to cognitive maps describing the structure of many different tasks (Eichenbaum and Cohen, 2014; Tavares et al., 2015; Solomon et al., 2019). […] Our study explored whether the isoto-allocortical axis also reflects a large-scale functional motif, and whether the axis interactions could help understand how the MTL elicits a broad role in neural function.”

2) Some of the methods are not fully described and are hard to understand. For example, the surface models that are used to sample and model the properties of the microstructure at different cortical depths could be described in more detail. I was also having trouble understanding two things about the "confluence" or "intersection" between the allocentric and isocentric cortices. I was left wondering if the intersection is defined as a plane in surface space, demarcating the separation between hippocampus and entorhinal cortex? Is the confluence/intersection defined based on the manual hippocampal subfields (i.e. medial boundary of the subiculum) or is it defined some other way using the surface profiles/features? Finally, how is geodesic "distance" computed? I would suggest adding a figure to give an overview of these aspects of the methods.

We thank the reviewer for pointing out these unclear aspects. Our continuous surface model was created by bridging isocortical and hippocampal surface meshes, so the intersection represents the new triangles in the surface mesh that link extant isocortical and hippocampal models. The position of the intersection was defined as the vertices with a minimum value on proximal-distal axis, which was labelled as the medial border of subiculum (DeKraker et al., 2019). Additionally, geodesic distance was calculated as the Chamfer distance along the surface mesh from the intersection. We have detailed the procedure in the updated Materials and methods and added a new Figure 5 to aid comprehension:

“An ultra-high resolution Merker stained 3D volumetric histological reconstruction of a post mortem human brain from a 65-year-old male was obtained from the open-access BigBrain repository, along with pial and white matter surface reconstructions (Amunts et al., 2013) (Figure 5A). […] The entorhinal, parahippocampal, and fusiform areas were defined using the Desikan-Killany atlas, nonlinearly transformed to the BigBrain histological surfaces (Lewis, Lepage and Evans, 2019), and the hippocampal subfields were assigned in line with manual segmentation (DeKraker et al., 2019). Only the right hemisphere was investigated, owing to a tear in the left entorhinal cortex in BigBrain.”

3) Related to the point above, I get the impression that these data show there is no strict boundary between the allo and iso-cortex but rather that there is a somewhat smooth gradient. This point could be made more clear in the Abstract and Discussion. What implications does this particular finding have for theories of MTL subregion function?

Indeed, several prior researchers have suggested that the cytoarchitectural transitions between these cortical types are gradual and smooth (Sanides, 1970; Braak and Braak, 1985; Insausti et al., 2017). Our data-driven approach supported the graded nature of the transition. In terms of MTL subregions, our findings link signal flow through hippocampal subfields to this cytoarchitectural gradient and provide a complementary framework to understand functions that are dissociated by subfields along this axis, such pattern separation/completion (Stevenson et al., 2020) and encoding/retrieval (Suthana et al., 2015). In addition, by showing functional connectivity variations along iso-to-allocortical and long-axes to previously established large-scale gradients and community organization, we demonstrate how such differences can be related to large-scale modes of brain function – specifically multiple demand networks representing different task states and sensory-transmodal hierarchies showing a dissociation between external and internally generated functional systems. We’ve expanded upon this issue in the Discussion and rephrased the Abstract and Discussion to emphasise the continuity of the iso-to-allocortical axis.

“exhibits a gradual cytoarchitectural transition from six-layered parahippocampal isocortex to three-layered hippocampal allocortex.”

“To this end, we integrated manual hippocampal segmentation (DeKraker et al., 2019) with an isocortical surface to create a smooth, continuous model on an ultra-high resolution 3D histological reconstruction of the human brain (Amunts et al., 2013).”

“To our knowledge, no study has specifically examined continuous variations in functional connectivity along the isoto-allocortical axis of the MTL; however, our results align with observed participation of the parahippocampus and fusiform gyri in externally-oriented and attention networks (Andrews-Hanna et al., 2010; Yeo et al., 2011; Qin et al., 2016). The iso-to-allocortical axis also mirrors a reduction in the diversity of projections (Lavenex and Amaral, 2000), which further explains the shift from involvement in large-scale polysynaptic networks toward more constrained, local connectivity. These findings may help to understand the influence of cytoarchitecture and connectivity on functional dissociation within the MTL, as subfield-focused approaches have shown selective task activation along the iso-toallocortical axis, for example during pattern separation/completion (Stevenson et al., 2020) and encoding/retrieval (Suthana et al., 2015).”

4) When r-values are reported to differ for different gradients (e.g. iso versus allo) it is important to test for a significant difference in the slopes (e.g. Fisher r-to-z transform or similar) to know if the relationships are statistically different from one another.

In the updated manuscript, we assessed the statistical difference of correlation coefficients using an extension of the Fisher r-to-z transform that takes into account shared and correlated variables (Meng, Rosenthal and Rubin, 1992). We have updated the Materials and methods and Results accordingly:

“We further stratified these edges by the direction along the axis (i.e., signal flow towards isocortex or towards allocortex) and compared the correlation coefficients in each direction (Meng, Rosenthal and Rubin, 1992).”

“Deviation from the iso-to-allocortical axis was related to lower effective connectivity in both directions (to isocortex r_iso_=-0.43, p=0.02; to allocortex: r_allo_=-0.44, p=0.02; difference in coefficients: z=-4.33, p=0.99, Figure 2B right).”

“Unlike the full mesiotemporal confluence, however, we did not consistently observe inverse correlation between deviation from the iso-to-allocortical axis and effective connectivity in these spatially restricted models (difference in coefficients: anterior z=-2.92, p=0.99; middle z=-1.01, p=0.84; posterior z=-0.47, p=0.68, Figure 2C).”

“Again, the association was specific to one gradient, with higher correlation coefficients of the anterior-posterior axis with the first functional gradient than the second (z=19.21, p<0.001) and third (z=30.23, p<0.001).”

5) This paper builds nicely on other work by DeKraker and colleagues (2019) that has analyzed the microstructural properties of the hippocampus. I think the readers of this paper would appreciate a brief description of how this investigation is similar/different from that work. For example, are the "features" identified here largely overlapping with those identified by DeKraker, and if not, how do they differ here?

We thank the reviewer for suggesting further contextualization of the work of DeKraker et al., 2019, and to more clearly outline similarities and differences from this work. We have added the following to the Introduction and Discussion:

“Nevertheless, compared to other mammals, the human MTL has exaggerated folding, more extensive lamination of the entorhinal cortex as well as a more prominent appearance of the Cornu Ammonis (CA) 2 subfield, highlighting the need to characterise the region’s microstructure in our species (Insausti and Amaral, 2012; Duvernoy et al., 2013; Insausti et al., 2017). Previous studies have detailed the distinct cytoarchitectural properties of hippocampal subfields (DeKraker et al., 2019), but cytoarchitectural patterns that extend across the subfields and their relation to the isocortex are less clear.”

“We characterised microstructure profiles by the central moments, shown to discriminate between specific MTL segments (DeKraker et al., 2019), as well as depth-wise intensities that provide a complementary and more direct interpretation.”

6) In the effective connectivity analysis of the MTL, how is variability of the MTL anatomy taken into account? For example, the fusiform and parahippocampal regions of interest will contain highly variable anatomical structures across subjects (e.g. different folding patterns of the collateral sulcus). Given that the focus on anatomical specificity is a major strength of this paper, I would be curious to know how anatomical variability/specificity is accounted for when the data are morphed into MNI152 volume space.

We tried to optimise the mapping of the histologically-defined axis to individual MRIs by using nonlinear transformations between MNI152 space and the native subject space, then masking the transformed atlas with a native space grey matter mask (derived from the T1w image). Of note, in the previous manuscript we mistakenly skipped these steps in our description of the fMRI preprocessing and have clarified this in the updated Materials and methods.

“Finally, for each subject, we generated a nonlinear transformation (MNI152 to native rs-fMRI space)(Andersson, Jenkinson and Smith, 2007) and a binary grey matter mask (>50% tissue-probability in T1w image) to facilitate mapping the standardised volumetric atlas to individual subjects’ functional images.”

To further benchmark how far this parcel mapping was consistent across individuals, we calculated the product-moment correlation coefficient of rs-fMRI timeseries within bins of the iso-to-allocortical axis transformed to native subject space. We found higher functional homogeneity within bins than between across our sample, indicating that the mapping of the axis to individual anatomy was functionally consistent across individuals. We’ve expanded upon these issues in the Materials and methods.

“Additionally, to assess the mapping of the iso-to-allocortical axis to individual fMRI scans, we calculated the functional homogeneity of rs-fMRI timeseries within bins in native subject space then averaged across subjects (Figure 2—figure supplement 1B). Across individuals, we were able to show higher functional homogeneity within bins than between, in line with the range of a previous study using individualised structural segmentation (Zhong et al., 2019), supporting the accurate mapping of the axis to individual anatomy.”

Nevertheless, as mentioned in the Limitations, it would be optimal in future work to define the isoto-allocortical axis within each subject separately:

"Follow up work based on ultra-high field magnetic resonance imaging (Yushkevich et al., 2006; DeKraker et al., 2018) may take similar approaches to explore MTL transitions with respect to in vivo cortical myeloarchitecture or intrinsic signal flow across cortical depths (Polimeni et al., 2010; Huber et al., 2017; Finn et al., 2019; Paquola et al., 2019), thus allowing assessment of how inter-individual differences in structural and functional organisation of the MTL relate to inter-individual differences in cognitive and affective phenotypes. In addition, this would allow replication in the left hemisphere, which exhibits nuanced cytoarchitectural differences to the right MTL (Zaidel, 1999; Barrera et al., 2001) and asymmetry in task activations (Shipton et al., 2014; Miller et al., 2018). In the present work, we tried to optimise alignment to in vivo fMRI using nonlinear registration and evaluated the accuracy against individual functional anatomy (Figure 2—figure supplement 1B)”

7) I was unsure which analyses were replicated in the Human Connectome Project (HCP) dataset. It is stated that the isocortical functional gradients were re-generated within the HCP cohort and that results were "highly similar" to the original dataset. Was this similarity formally tested?

We found that functional gradients derived in the original MICs and replication HCP datasets were highly similar based on Spearman correlations, adjusted for spatial autocorrelations using spin tests (r_G1_=0.78, p_spin_<0.001, r_G3_=0.81, p_spin_<0.001). We have added this analysis to the Materials and methods:

“The isocortical functional gradients were re-generated within the HCP cohort and were highly similar to the original dataset (Spearman correlation between datasets: r_G1_=0.78, p_spin_<0.001, r_G3_=0.81, p_spin_<0.001).”

We have also clarified the tests carried out in the replication dataset in the updated Materials and methods:

“Specifically, we tested whether effective connectivity was associated with the deviation from the iso-to-allocortical axis, in the whole MTL and the anterior-posterior divisions, whether rs-fMRI along the iso-to-allocortical axis was spatially correlated with the multiple demand functional gradient and whether rs-fMRI along the anterior-posterior axis was spatially correlated with the sensory-transmodal functional gradient.”

Reviewer #3:Substantive concerns:This is an interesting study which shows novel relationships between mesial temporal structures and whole brain functional organization. As the authors point out, the novel part of the study involves defining cytoarchitectural regions, and correlating these changes with both local and global function as defined by BOLD fMRI. This is a novel study examining the iso-allocortical transitions with the MTL, and correlating them with local and global rs-fMRI changes. As the authors state, the global rs-fMRI findings related the anterior-posterior axis of the MTL are not new, but add complimentary findings in comparison the iso-allocortical transition findings. Given this, I will focus my comments on the use of the BigBrain image, and definition of the MTL transitions for use in defining regions in the rs-fMRI images.1) With the BigBrain data, only the right hippocampus was used for segmentation, due to a rip in the histopathological sections of entorhinal cortex on the left. It is therefore assumed that the right MTL segmentations were inverted and also used for the left MTL rs-fMRI analysis. If this is the case, it should be more clearly stated in the Materials and methods. Also, discussion should be added to the possible implications for results, both in respect to replicating the histological intensity findings (which could be tested in two hippocampi if both right and left were processed) and the known structural differences between the right and left hippocampi.

We thank the reviewer for the comment. All analyses were indeed conducted using only the right MTL, in other words no mirroring procedure was implemented. We clarified this in the revised Materials and methods.

“Only the right hemisphere was investigated, owing to a tear in the left entorhinal cortex in BigBrain.”

“Notably, only the right MTL was modelled and all subsequent analyses relate to the right MTL.”

In addition, we also extended upon this issue in the Discussion:

“Follow up work based on ultra-high field magnetic resonance imaging may take similar approaches to explore MTL transitions with respect to in vivo cortical myeloarchitecture or intrinsic signal flow across cortical depths (Polimeni et al., 2010; Huber et al., 2017; Finn et al., 2019; Paquola et al., 2019), thus allowing assessment of how interindividual differences in structural and functional organisation of the MTL relate to inter-individual differences in cognitive and affective phenotypes. In addition, this would allow replication in the left hemisphere, which exhibits nuanced cytoarchitectural differences to the right MTL (Zaidel, 1999; Barrera et al., 2001) and asymmetry in task activations (Shipton et al., 2014; Miller et al., 2018).”

2) I had concerns that using the higher resolution BigBrain image as a template for the 8 nodes in the MTL for the much lower resolution rs-fMRI images would be problematic for signal to noise ratio. However, the authors have convincingly shown consistent findings when controlling for signal to noise ratios.

We agree with the reviewer and tried to address this concern by a) establishing an accurate mapping of the histologically-defined iso-to-allocortical axis to rs-fMRI using nonlinear transformation and b) by accounting for signal to noise ratio in our analyses.

To further assess the accuracy of the transformation, we calculated the product moment correlation coefficient of rs-fMRI timeseries within bins of the iso-to-allocortical axis transformed to native subject space. We found higher functional homogeneity within bins than between, supporting accurate mapping of the axis to individual anatomy. We have expanded upon these issues in the revised Materials and methods:

“Additionally, to assess the mapping of the iso-to-allocortical axis to individual fMRI scans, we calculated the functional homogeneity of rs-fMRI timeseries within bins (Figure 2—figure supplement 1B). We were able to show higher functional homogeneity within bins than between, line with the range of a previous study using individualised structural segmentation (Zhong et al., 2019), supporting the accurate mapping of the axis to individual anatomy.”

Additionally, we expanded upon this issue in the Limitations:

"Follow up work based on ultra-high field magnetic resonance imaging (Yushkevich et al., 2006; DeKraker et al., 2018) may take similar approaches to explore MTL transitions with respect to in vivo cortical myeloarchitecture or intrinsic signal flow across cortical depths (Polimeni et al., 2010; Huber et al., 2017; Finn et al., 2019; Paquola et al., 2019), thus allowing assessment of how inter-individual differences in structural and functional organisation of the MTL relate to inter-individual differences in cognitive and affective phenotypes. In addition, this would allow replication in the left hemisphere, which exhibits nuanced cytoarchitectural differences to the right MTL (Zaidel, 1999; Barrera et al., 2001) and asymmetry in task activations (Shipton et al., 2014; Miller et al., 2018). In the present work, we tried to optimise alignment to in vivo fMRI using nonlinear registration and evaluated the accuracy against individual functional anatomy (Figure 2—figure supplement 1B)”

3) The authors mention (and reference) the correlation of histopathological cellular staining intensities with cellular densities and soma size in the Materials and methods section. Given the centrality of this concept to their findings of the BigBrain data, some addition to the discussion about this concept and the underlying evidence for correlation of staining intensity and cellular densities and soma size would be helpful.

We’ve expanded upon the description of the staining procedure in the revised Materials and methods:

“The Merker staining is a form of silver impregnation for cell bodies that produces a high contrast of black pigment in cells on a virtually colorless background (Merker, 1983). Stained histological sections were then digitized at 20µm, resulting in greyscale images with darker colouring where many or large cells occur. The density and size of cells varies across laminar, as well as areas, thus capturing the regional differentiation of cytoarchitecture.”